# Identification, Amplification and Measurement: A bridge to Gaussian Differential Privacy

**Yi Liu**
Department of Mathematical
and Statistical Sciences
University of Alberta
yliu16@ualberta.ca

**Ke Sun**
Department of Mathematical
and Statistical Sciences
University of Alberta
ksun6@ualberta.ca

**Bei Jiang**
Department of Mathematical
and Statistical Sciences
University of Alberta
bei1@ualberta.ca

**Linglong Kong**
Department of Mathematical
and Statistical Sciences
University of Alberta
lkong@ualberta.ca

## Abstract

Gaussian differential privacy (GDP) is a single-parameter family of privacy notions that provides coherent guarantees to avoid the exposure of sensitive individual information. Despite the extra interpretability and tighter bounds under composition GDP provides, many widely used mechanisms (e.g., the Laplace mechanism) inherently provide GDP guarantees but often fail to take advantage of this new framework because their privacy guarantees were derived under a different background. In this paper, we study the asymptotic properties of privacy profiles and develop a simple criterion to identify algorithms with GDP properties. We propose an efficient method for GDP algorithms to narrow down possible values of an optimal privacy measurement, $\mu$ with an arbitrarily small and quantifiable margin of error. For non GDP algorithms, we provide a post-processing procedure that can amplify existing privacy guarantees to meet the GDP condition. As applications, we compare two single-parameter families of privacy notions, $\epsilon$-DP, and $\mu$-GDP, and show that all $\epsilon$-DP algorithms are intrinsically also GDP. Lastly, we show that the combination of our measurement process and the composition theorem of GDP is a powerful and convenient tool to handle compositions compared to the traditional standard and advanced composition theorems.

## 1 Introduction

Recent years have seen explosive growth in the research and application of data-driven machine learning. While data fuels advancement in this unprecedented age of "big data", concern for individual privacy has deepened with the continued mining, transportation, and exchange of this new resource. While expressions of privacy concerns can be traced back as early as 1969 [27], the concept of privacy is often perceived as "vague and difficult to get into a right perspective" [34]. Through its alluring convenience and promise of societal prosperity, the use of aggregated data has long outstripped the capabilities of privacy protection measures. Indeed, early privacy protection protocols relied on the ad hoc enforcement of anonymization and offered little to no protection against the exposure of individual data, as evidenced by the AOL search log and Netflix Challenge dataset controversies [30, 31, 6].

36th Conference on Neural Information Processing Systems (NeurIPS 2022).

Differential privacy (DP) first gained traction as it met the urgent need for rigour and quantifiability in privacy protection [14]. In short, DP bounds the change in the distribution of outputs of a query made on a dataset under an alteration of one data point. The following definition formalizes this notion.

**Definition 1.1** *[14] A randomized algorithm $\mathcal{A}$, taking a dataset consisting of individuals as its input, is $(\epsilon, \delta)$-differentially private if, for any pair of datasets $S$ and $S'$ that differ in the record of a single individual and any event $E$,*

$$P[\mathcal{A}(S) \in E] \leq e^\epsilon P\left[\mathcal{A}\left(S'\right) \in E\right] + \delta.$$

*When $\delta = 0$, $\mathcal{A}$ is called $\epsilon$-differentially private ($\epsilon$-DP).*

While the notion of $(\epsilon, \delta)$-DP has wide applications [12, 16, 10, 21], there are a few notable drawbacks to this framework. One is the poor interpretability of $(\epsilon, \delta)$-DP: unlike other concepts in machine learning, DP should not remain a black box. Privacy guarantees are intended for human interpretation and so must be understandable by the users it affects and by regulatory entities. A second drawback is $(\epsilon, \delta)$-DP's inferior composition properties and lack of versatility. Here, "composition" refers to the ability for DP properties to be inherited when DP algorithms are combined and used as building blocks. As an example, the training of deep learning models involves gradient evaluations and weight updates: each of these steps can be treated as a building block. It is natural to expect that a DP learning algorithm can be built using differentially-private versions of these components. However, the DP composition properties cannot generally be well characterized within the framework of $(\epsilon, \delta)$-DP, leading to very loose composition theorems.

To overcome the drawbacks of $(\epsilon, \delta)$-DP, numerous variants have been developed, including the hypothesis-testing-based $f$-DP [35, 13], the moments-accountant-based Rényi DP [28], as well as concentrated DP and its variants [9, 8]. Despite their very different perspectives, all of these DP variants can be fully characterized by an infinite union of $(\epsilon, \delta)$-DP guarantees. In particular, there is a two-way embedding between $f$-DP and the infinite union of $(\epsilon, \delta)$-DP guarantees: any guarantee provided by an infinite union of $(\epsilon, \delta)$-DP can be fully characterized by $f$-DP and vice visa [13]. Consequently, $f$-DP has the versatility to treat all of the above notions as special cases.

In addition to its versatility, $f$-DP is more interpretable than other DP paradigms because it considers privacy protection from an attacker's perspective. Under $f$-DP, an attacker is challenged with the hypothesis-testing problem

$$H_0 : \text{ the underlying dataset is } S \qquad \text{versus} \qquad H_1 : \text{ the underlying dataset is } S'$$

and given output of an algorithm $\mathcal{A}$, where $S$ and $S'$ are neighbouring datasets. The harder this testing problem is, the less privacy leakage $\mathcal{A}$ has. To see this, consider the dilemma that the attacker is facing. The attacker must reject either $H_0$ or $H_1$ based on the given output of $\mathcal{A}$: this means the attacker must select a subset $R_0$ of $\text{Range}(\mathcal{A})$ and reject $H_0$ if the sampled output is in $R_0$ (or must otherwise reject $H_1$). The attacker is more likely to incorrectly reject $H_0$ (in a type I error) when $R_0$ is large. Conversely, if $R_0$ is small, the attacker is more likely to incorrectly reject $H_1$ (in a type II error). We say that an algorithm $\mathcal{A}$ is $f$-DP if, for any $\alpha \in [0, 1]$, no attacker can simultaneously bound the probability of type I error below $\alpha$ and bound the probability of type II error below $f(\alpha)$. Such $f$ is called a trade-off function and controls the strength of the privacy protection.

The versatility afforded by $f$ can be unwieldy in practice. Although $f$-DP is capable of handling composition and can embed other notions of differential privacy, it is not convenient for representing safety levels as a curve amenable to human interpretation. Gaussian differential privacy (GDP), as a parametric family of $f$-DP guarantees, provides a balance between interpretability and versatility. GDP guarantees are parameterized by a single value $\mu$ and use the trade-off function $f(\alpha) = \Phi\left(\Phi^{-1}(1 - \alpha) - \mu\right)$, where $\Phi$ is the cumulative distribution function of the standard normal distribution. With this choice of $f$, the hypothesis-testing problem faced by the attacker is as hard as distinguishing between $N(0, 1)$ and $N(\mu, 1)$ on the basis of a single observation. Aside from its visual interpretation, GDP also has unique composition theorems: the composition of a $\mu_1$- and $\mu_2$-GDP algorithm is, as expected, $\sqrt{\mu_1^2 + \mu_2^2}$-GDP. This property can be easily generalized to $n$-fold composition. GDP also has a special central limit theorem implying that all hypothesis-testing-based definitions of privacy converge to GDP in terms of a limit in the number of compositions. Readers are referred to [13] for more information.

## 1.1 Outline

The goal of this paper is to provide a bridge between GDP and algorithms developed under other DP frameworks. We start by presenting an often-overlooked partial order on $(\epsilon, \delta)$-DP conditions induced by logical implication. Ignoring this partial order will lead to problematic asymptotic analysis.

We then break down GDP into two parts: a head condition and a tail condition. We show that the latter, through a single limit of a mechanism's privacy profile, is sufficient to distinguish between GDP and non-GDP algorithms. For GDP algorithms, this criterion also provides a lower bound for the privacy protection parameter $\mu$ and can help researchers widen the set of available GDP algorithms. This criterion furthermore gives an interesting characterization of GDP without an explicit reference to the Gaussian distribution.

The next logical step is to measure the exact privacy performance. Interestingly, while the binary "GDP or not" question can be answered solely by the tail, the actual performance of a DP algorithm is determined by the head. We define and apply the Gaussian Differential Privacy Transformation (GDPT) to narrow the set of potential optimal values of $\mu$ with an arbitrarily small and quantifiable margin of error. We further provide procedure to adapt an algorithm to GDP or improve the privacy parameter when results from the GDP identification and measurement procedures are undesirable.

Lastly, we demonstrate additional applications of our newly developed tools. We first make a comparison between DP and GDP and show that any $\epsilon$-DP algorithm is automatically GDP. We then show that the combination of our measurement process and the GDP composition theorem is a more powerful and convenient tool for handling compositions relative to traditional composition theorems.

## 2 Privacy profiles and an exact partial order on $(\epsilon, \delta)$-DP conditions

The benefits of DP come with a price. As outlined in the definition of DP, any DP algorithm must be randomized. This randomization is usually achieved by perturbing the intermediate step or the final output via the injection of random noise. Because of the noise, a DP algorithm cannot faithfully output the truth like its non DP counterpart. To provide a higher level of privacy protection, a stronger utility compromise should be made. This leads to the paramount problem of the "privacy–utility trade-off". Under the $(\epsilon, \delta)$-DP framework, this trade-off is often characterized in a form of $\sigma = f(\epsilon, \delta)$: to achieve $(\epsilon, \delta)$-DP, the utility parameter (usually the scale of noise) needs to be chosen as $f(\epsilon, \delta)$. Therefore, an algorithm can be $(\epsilon, \delta)$-DP for multiple pairs of $\epsilon$ and $\delta$: the union of all such pairs provides a complete image of the algorithm under the $(\epsilon, \delta)$-DP framework. In particular, an $(\epsilon, \delta)$-DP mechanism $\mathcal{A}$ is also $(\epsilon', \delta')$-DP for any $\epsilon' \geq \epsilon$ and any $\delta' \geq \delta$. The infinite union of $(\epsilon, \delta)$ pairs can thus be represented as the smallest $\delta$ associated with each $\epsilon$. This intuition is formulated as a privacy profile in [5]. The privacy profile corresponding to a collection of $(\epsilon, \delta)$-DP guarantees $\Omega$ is defined as the curve in $[0, \infty) \times [0, 1]$ separating the space of privacy parameters into two regions, one of which contains exactly the pairs in $\Omega$. The privacy profile provides as much information as $\Omega$ itself. Many privacy guarantees and privacy notions, including $(\epsilon, \delta)$-DP, Rényi DP, $f$-DP, GDP, and concentrated DP, can be embedded into a family of privacy profile curves and fully characterized [3]. A privacy profile can be provided or derived by an algorithm's designer or users.

Before proceeding with detailed discussions, we first give three examples of DP algorithms that are used throughout the paper. The first example we consider is the Laplace mechanism, a classical DP mechanism whose prototype is discussed in the paper that originally defined the concept of differential privacy [14]. The level of privacy that the Laplace mechanism can provide is determined by the scale $b$ of the added Laplacian noise. Given a global sensitivity $\Delta$, the value of $b$ needs to be chosen as $f(\epsilon, 0) = \Delta/\epsilon$ in order to provide an $(\epsilon, 0)$-DP guarantee. Despite its long history, the Laplace mechanism has remained in use and study in recent years [32, 22, 36, 25]. Our second example is a family of algorithms in which a noise parameter has the form $\sigma = A\epsilon^{-1}\sqrt{\log(B/\delta)}$. Examples include: the goodness of fit algorithm [18], noisy stochastic gradient descent and its variants [7, 1, 17] and the one-shot spectral method and the one-shot Laplace algorithm [33]. Our third example comes from the field of federated learning: given $n$ users and the number of messages $m$, the invisibility cloak encoder algorithm (ICEA) from [23] is $(\epsilon, \delta)$-DP if $m > 10 \log(n/(\epsilon\delta))$ [20]. See also [4, 19] for other analysis of ICEA.

For figures and numerical demonstrations in this paper, we use $b = 2/\Delta$ for the Laplace mechanism; $A = 2$, $B = 1$, and $\sigma = 2$ for the second example, which we refer to as SGD; and $m = 20$

and $n = 4$ for the ICEA. We omit the internal details of these methods and focus on their privacy guarantees: other than for the classical Laplace mechanism, whose privacy profile is known [3], privacy guarantees are given in the form of a privacy–utility trade-off equation $\sigma = g(\epsilon, \delta)$. Given $\sigma$, it is tempting to derive the privacy profile by inverting $g$ (i.e., as $\delta_{\mathcal{A}}(\epsilon) = \min\{\delta \mid \sigma = g(\epsilon, \delta)\}$) because an $(\epsilon_0, \delta_0)$-DP algorithm is trivially $(\epsilon, \delta)$-DP for any $\epsilon \geq \epsilon_0$ and $\delta \geq \delta_0$. However, in most cases, a privacy profile naively derived in this way is not tight and will lead to a problematic asymptotic analysis, especially near the origin, because of a frequently overlooked partial order between $(\epsilon, \delta)$-DP conditions below.

**Theorem 2.1** *Assume that $\epsilon_0 \geq 0$ and $0 \leq \delta_0 < 1$. The $(\epsilon_0, \delta_0)$-DP condition implies $(\epsilon, \delta)$-DP if and only if $\delta \geq \delta_0 + (1 - \delta_0)(e^{\epsilon_0} - e^{\epsilon})^+/(1 + e^{\epsilon_0})$.*

Theorem 2.1 states the exact partial order of logical implication on $(\epsilon, \delta)$-DP conditions. Though not being explicitly discussed in this form in previous literature on DP, this partial order can be implicitly derived from other results (e.g. proposition 2.11 of [13]). Taking this partial order into account, the privacy profile derived from the naive inversion of the trade-off function can be refined into

$$\delta_{\mathcal{A}}(\epsilon) = \min\left( \left\{ \delta \mid \sigma = g(\epsilon_0, \delta_0) \text{ and } \delta \geq \delta_0 + \frac{(1 - \delta_0)(e^{\epsilon_0} - e^{\epsilon})^+}{1 + e^{\epsilon_0}} \right\} \right).$$

Intuitively, the refined privacy profile not only considers $(\epsilon, \delta)$-DP provided directly by the trade-off function but also takes all pairs $(\epsilon, \delta)$ inferred by corollary 2.1. See figure 1 for comparison before and after this refinement.

## 3 The identification of GDP algorithms

We next show the connection between GDP and the privacy profile: briefly, Gaussian differential privacy can be characterized as an infinite union of $(\epsilon, \delta)$-DP conditions.

**Theorem 3.1** *([Corollary 2.13 [13]) A mechanism is $\mu$-GDP if and only if it is $(\epsilon, \delta_\mu(\epsilon))$-DP for all $\epsilon \geq 0$, where*

$$\delta_\mu(\epsilon) = \Phi\left( -\frac{\epsilon}{\mu} + \frac{\mu}{2} \right) - \mathrm{e}^\epsilon \Phi\left( -\frac{\epsilon}{\mu} - \frac{\mu}{2} \right). \tag{1}$$

This result follows from properties of $f$-DP. Prior to this general form, a expression for a special case appeared in [5]. From the definition of the privacy profile, it follows immediately that an algorithm $\mathcal{A}$ with the privacy profile $\delta_{\mathcal{A}}$ is $\mu$-GDP if and only if $\delta_\mu(\epsilon) \geq \delta_{\mathcal{A}}(\epsilon)$ for all non-negative $\epsilon$. However, this observation does not automatically lead to a meaningful way to identify GDP algorithms.

Before proceeding with an analysis of privacy profiles, we give a few visual examples in Figure 1. The left side of1 illustrates the privacy profiles of our examples. That of the Laplace mechanism is derived in [3] as Theorem 3: given a noise parameter $b$ and a global sensitivity $\Delta$, the privacy profile of the Laplace mechanism is $\delta(\epsilon) = \max(1 - \exp\{\varepsilon/2 - \Delta/(2b)\},\ 0)$. For the second and the third examples, we compare the naive privacy profiles obtained by inverting the trade-off function with the refined privacy profiles. The refined and naive privacy profiles take on notably different values around $\epsilon = 0$. The inverted trade-off functions suggest that $(0, \delta)$ cannot be achieved by any choice of parameter $\sigma$. However, this is clearly not true, considering Theorem 2.1.

As shown in right side of Figure 1, the Laplace mechanism's privacy profile is below the 2-GDP and 4-GDP curves but crosses the 1-GDP curve, indicating that the Laplace mechanism in this case is 2-GDP and 4-GDP but not 1-GDP. The ICEA curve intersects all of the displayed GDP curves, so the algorithm is not $\mu$-GDP for $\mu \in \{1, 2, 4\}$. It is hard to tell whether or not the SGD curve crosses the 1-GDP curve and we cannot say if it will cross the 2-GDP or even the 4-GDP curve at a large value of $\epsilon$. These examples illustrate that we cannot draw conclusions simply by looking at a graph. A privacy profile is defined on $[0, \infty)$, so it is hard to tell if an inequality is maintained as $\epsilon$ increases. Previous failures of ad hoc attempts at privacy have taught that privacy must be protected via tractable and objective means [30, 31, 6].

Performing this check via numerical evaluation yields similar problems: we cannot consider all values of $\epsilon$ on an infinite interval (or even a finite one, for that matter). Turning to closed forms for privacy profiles and $\delta_\mu$ is also difficult: even if a given privacy profile is easy to handle, $\delta_\mu$ presents some technical hurdles. The profile $\delta_\mu$ and $\Phi$ are transcendental with different asymptotic behaviors for

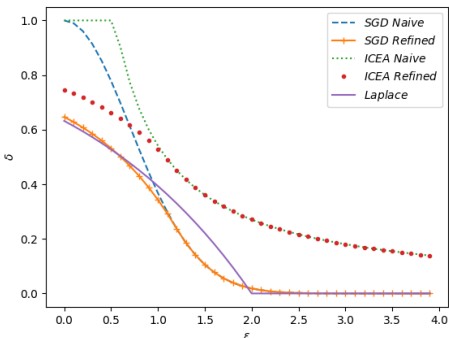 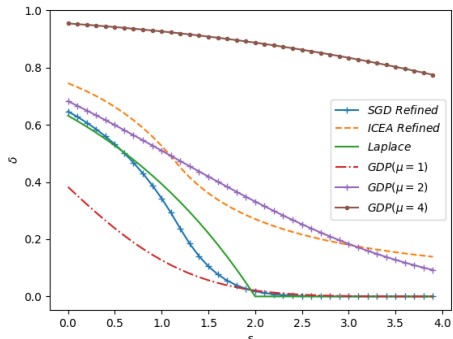

Figure 1: Left: Examples of privacy profiles obtained by inverting the trade-off function (naive) and by Theorem 2.1 (refined). Right: Comparison of 1-GDP and 2-GDP privacy profiles against those for our three examples.

different values of $\mu$ and $\epsilon$. This is clear from the Figure 1: near $\epsilon = 0$, $\delta_\mu$ is concave for $\mu = 4$ but convex for $\mu = 1$. As a further complication, both the first and second terms in the definition of $\delta_\mu$ converge to 1 as $\epsilon \to \infty$, but the difference between them vanishes. Subtracting good approximations of two nearby numbers may cause a phenomenon called catastrophic cancellation and lead to very bad approximations [26, 11]. Due to the risk of catastrophic cancellation, a good approximation of $\Phi$ does not guarantee a good approximation of the GDP privacy profile. These problems make it difficult to tightly bound $\delta_\mu$ by a function with a simple form.

To address the problem of differing asymptotic behaviours, we define the following two notions.

**Definition 3.1** *(Head condition) An algorithm $\mathcal{A}$ with the privacy profile $\delta_{\mathcal{A}}$ is $(\epsilon_h, \mu)$-head GDP if and only if $\delta_{\mathcal{A}}(\epsilon) \leq \delta_\mu(\epsilon)$ when $\epsilon \leq \epsilon_h$.*

**Definition 3.2** *(Tail condition) An algorithm $\mathcal{A}$ with the privacy profile $\delta_{\mathcal{A}}$ is $(\epsilon_t, \mu)$-tail GDP if and only if $\delta_{\mathcal{A}}(\epsilon) \leq \delta_\mu(\epsilon)$ when $\epsilon > \epsilon_t$.*

The head condition checks the $\mu$-GDP condition for $\epsilon$ near zero and the tail condition checks the $\mu$-GDP condition for $\epsilon$ far away from zero. As such, the combination of $(\epsilon, \mu)$-head GDP and $(\epsilon, \mu)$-tail GDP is equivalent to $\mu$-GDP. For now, we put the exact value of $\mu$ aside and consider only the qualitative question of how to identify a GDP algorithm by its privacy profile. The following theorem answers this question.

**Theorem 3.2** *An algorithm $\mathcal{A}$ is GDP if and only if $\mathcal{A}$ is $(\epsilon, \mu)$-tail GDP for any finite $\epsilon$ and $\mu$.*

Interestingly, only the tail condition figures into the identification problem. The reason for this stems from theorem 2.1. Any nontrivial $(\epsilon, \delta)$-DP algorithm must be $(0, \delta)$-DP for some $\delta < 1$ and therefore must satisfy a head condition for some sufficiently large $\mu$. The only problem left is the tail. However, it is not possible to check whether $\delta(\epsilon) < \delta_\mu(\epsilon)$ for all values of $\epsilon$. To circumvent this issue, we present a key lemma that underlies much of the theoretical analysis in this section and may continue to be useful in future developments.

**Lemma 3.1** *Define $\tilde{\delta}_\mu(\epsilon) = \frac{\mu e^{-a^2/2}}{\sqrt{2\pi}a^2}$, where $a = -\frac{\epsilon}{\mu} + \frac{\mu}{2}$. It follows that $\lim_{\epsilon \to +\infty} \frac{\delta_\mu(\epsilon)}{\tilde{\delta}_\mu(\epsilon)} = 1$.*

Using the key lemma above, a condition for identifying GDP algorithms is simple to formulate:

**Theorem 3.3** *Let $\mu_t = \sqrt{\lim_{\epsilon \to +\infty} \frac{\epsilon^2}{-2 \log \delta_{\mathcal{A}}(\epsilon)}}$. An algorithm $\mathcal{A}$ with the privacy profile $\delta_{\mathcal{A}}(\epsilon)$ is $\mu$-GDP if and only if $\mu_t < \infty$ and $\mu$ is no smaller than $\mu_t$.*

Theorems 3.2 and 3.3 give a useful criterion characterizing GDP and deepen our understanding of GDP. Putting the exact value of $\mu$ aside, a GDP algorithm must provide an infinite union of $(\epsilon, \delta)$-DP conditions, where $\delta$ must be $O(e^{-\epsilon^2})$ as $\epsilon \to \infty$. Refer to Appendices B.3 for proofs of Theorems.

# 4 The Gaussian differential privacy transformation

While the binary "GDP or not" question can be answered solely by the tail condition, the actual performance of a DP algorithm is determined by the value of its privacy profile for small values of $\epsilon$: intuitively, all $(\epsilon_t, \mu)$-tail conditions are weaker than the corresponding $\epsilon_t$-DP condition, and the latter provides almost no privacy when $\epsilon_t > 10$. A more detailed discussion will be presented in 4.2. To solve the measurement problem, we first propose a new tool—the Gaussian differential privacy transformation (GDPT).

**Definition 4.1** *(GDPT) Let $f$ be a non-increasing, non-negative function defined on $[0, +\infty)$ satisfying $f(0) \leq 1$. The Gaussian differential privacy transformation (GDPT) of $f$ is the function $G_f$ mapping $[0, \infty)$ to $[0, \infty)$ such that $G_f(\epsilon) = \mu_{GDP}(\epsilon, f(\epsilon))$, where $\mu_{GDP}(x, y)$ is the implicit function defined by the equation $\delta_\mu(x) = y$.*

We highlight two critical features of the GDPT.

- The GDPT is order preserving: if $f(\epsilon) \geq g(\epsilon)$, then $G_f(\epsilon) \geq G_g(\epsilon)$.
- The GDPT of $\delta_\mu$ is $G_{\delta_\mu}(\epsilon) = \mu$, a constant function.

The first of these two features derive from the monotonicity of $\delta_\mu(\epsilon)$. Given a fixed $\mu$, $\delta_\mu(\epsilon)$ is a strictly decreasing continuous function of $\epsilon$. Given a fixed $\epsilon$, $\delta_\mu(\epsilon)$ is a strictly increasing continuous function of $\mu$. Therefore, $\mu_{GDP}(x, y)$ is an increasing function of $y$: this leads to the order-preserving property. The second property follows immediately from the definition of $\mu_{GDP}$.

By taking advantage of the order-preserving property, direct comparisons between $\delta_\mu$ and $\delta_{\mathcal{A}}$ are no longer necessary: instead, it is sufficient to compare their corresponding GDPTs. Furthermore, appealing to the second property above, we need only compare $G_{\mathcal{A}}$ to the constant function $\mu$. The following theorems formalize this insight.

**Corollary 4.1** *An algorithm $\mathcal{A}$ with the privacy profile $\delta_{\mathcal{A}}$ is $\mu$-GDP if and only if $\mu \geq \sup(\{G_{\mathcal{A}}(\epsilon) \mid \epsilon \in [0, \infty)\})$.*

**Theorem 4.1** *An algorithm $\mathcal{A}$ with the privacy profile $\delta_{\mathcal{A}}$ is $(\epsilon_h, \mu)$-head GDP or $(\epsilon_t, \mu)$-tail GDP if and only if $\mu \geq \sup(\{G_{\mathcal{A}}(\epsilon) \mid \epsilon \in [0, \epsilon_h]\})$ or $\mu \geq \sup(\{G_{\mathcal{A}}(\epsilon) \mid \epsilon \in (\epsilon_t, \infty)\})$, respectively.*

Without the above results, we would be forced to search through a large family of functions for a single $\delta_\mu$ that never crosses $\delta_{\mathcal{A}}$ anywhere on $[0, \infty)$ and has $\mu$ as small as possible. Now, with Theorem 4.1, we need only consider one function: the GDPT of $\delta_{\mathcal{A}}$. The tightest value $\mu$ is $\sup_\epsilon \{G_{\mathcal{A}}(\epsilon)\}$. Now we revisit our previous three examples for which the limit in Theorem 3.3 is 0, $\sqrt{1/2}$, and $+\infty$, respectively. From these evaluations, we can conclude that the Laplace mechanism and SGD are GDP and that the privacy profile of the ICEA algorithm crosses every $\mu$-GDP curve regardless of how large $\mu$ is, indicating that the ICEA algorithm is not GDP.

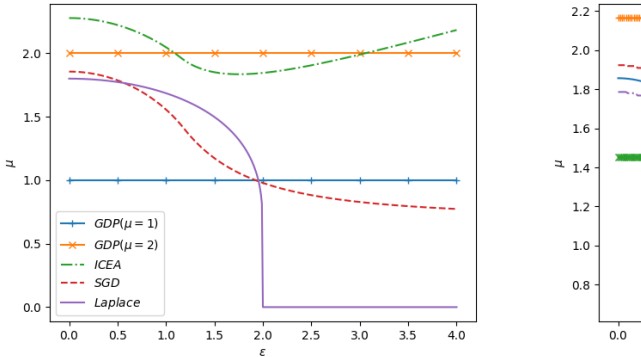 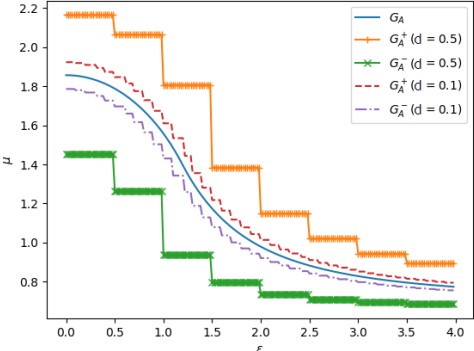

Figure 2: Left: Examples of GDPTs. Right: Plot of $G_{\mathcal{A}}^+$ and $G_{\mathcal{A}}^-$ with different values of $d$.

Left side of figure 2 shows the GDPTs of the three examples considered in this paper. All three GDPTs converge to a finite value as $\epsilon \to 0^+$. This can be attributed to the fact that any algorithm providing some non-trivial $(\epsilon, \delta)$-DP guarantee is $(0, \delta)$-DP for some $\delta \in [0, 1)$ (by theorem 2.1). For larger values of $\epsilon$, the GDPT of the Laplace mechanism takes on a constant value of 0, the GDPT of SGD converges to a value that is approximately 0.7, and the GDPT of the ICEA seems to be diverging. These observations are consistent with the values of 0, $\sqrt{1/2}$, and $\infty$ obtained from Theorem 3.3. Once an algorithm is confirmed to be GDP via Theorems 3.2 and 3.3, it is natural to be interested in the exact level of privacy protection, quantified by $\mu$. Nonetheless, plots are only good for visualization and are not sufficient proof when verifying GDP. We still need objective and tractable methods for obtaining bounds on GDPTs.

## 4.1 Measuring the head

Following the intuition outlined by definition 3.1 and 3.2, we decompose the GDP condition into head and tail conditions and first focus on finding $\mu$ such that $\mathcal{A}$ is $(\epsilon, \mu)$-head GDP. Without additional knowledge, finding $\sup\{G_{\mathcal{A}}(\epsilon) \mid \epsilon \in [0, \epsilon_h]\}$, even for a finite $\epsilon_h$, seems computationally infeasible. To solve this problem, we take advantage of the fact that $\mu_{\mathrm{GDP}}$ has a uniformly bounded partial derivative.

**Theorem 4.2** $0 \leq \frac{\partial \mu_{GDP}(\epsilon, \delta)}{\partial \epsilon} \leq \frac{\sqrt{2}\pi}{2}$.

The first half of the inequality above is no surprise to us: the GDP privacy measurement $\mu$ is expected to be larger when $\epsilon$ is larger. However, the second half allow us to only conduct the search on a finite list of $\epsilon$ without the concern of spikes in between. We formulate this insight as the following theorem:

**Theorem 4.3** *Given $\epsilon_h \geq 0$, let $d = \epsilon_h/n$ and $x_i = id$ for $i \in \{0, \ldots, n+1\}$. For $\epsilon \leq \epsilon_h$, the GDPT of $\mathcal{A}$, denoted by $G_{\mathcal{A}}(\epsilon)$, is bounded between the two staircase functions*

$$G_{\mathcal{A}}^-(\epsilon) = \sum_{i=0}^{n+1} \mu_{GDP}(x_i, \delta_{\mathcal{A}}(x_{i+1})) \times 1_{\epsilon \in [x_i, x_{i+1})} \ \ and \ G_{\mathcal{A}}^+(\epsilon) = \sum_{i=0}^{n+1} \mu_{GDP}(x_{i+1}, \delta_{\mathcal{A}}(x_i)) \times 1_{\epsilon \in [x_i, x_{i+1})}.$$

*Specifically,*

$$\max_{i \in \{0, \ldots, n\}} G_{\mathcal{A}}^-(x_i) \leq \max_{\epsilon \in [0, \epsilon_h]} G_{\mathcal{A}}(\epsilon) \leq \max_{i \in \{0, \ldots, n+1\}} G_{\mathcal{A}}^+(x_i) \leq \max_{i \in \{0, \ldots, n\}} G_{\mathcal{A}}^-(x_i) + \sqrt{2}\pi d. \tag{2}$$

Refer to Appendix B.5 and B.6 for proofs of Theorem 4.2 and 4.3, respectively.

For any $\epsilon_h < +\infty$, we can now bound any GDPT $G_{\mathcal{A}}$ to any precision on $[0, \epsilon_h]$ without full pointwise evaluation because $G_{\mathcal{A}}$ is bounded between $G_{\mathcal{A}}^+$ and $G_{\mathcal{A}}^-$ and each staircase function takes on only finitely many values. For any $c > 0$, the inequalities in (2) provide a viable way to bound $\max_{\epsilon \in [0, \epsilon_h]} G_{\mathcal{A}}(\epsilon)$ in an interval with a length no greater than $1/c$.

First, a binary search algorithm (algorithm 2 in Appendix D) can yield $\mu^+$ and $\mu^-$ such that $\mu^- \leq \mu_{\mathrm{GDP}}(\epsilon, \delta) \leq \mu^+$ and $\mu^+ - \mu^- < b$. For future references, we use $\mu_{\mathrm{GDP}}^+(\epsilon, \delta, b)$ and $\mu_{\mathrm{GDP}}^-(\epsilon, \delta, b)$ to represent such outputs of $\mu^+$ and $\mu^-$, respectively. Therefore, we can naively go thorough all $G_{\mathcal{A}}^-(x_i)$ and $G_{\mathcal{A}}^+(x_i)$. By picking $n = \lceil \sqrt{8}c\pi\epsilon_h \rceil + 1$ and $b = \frac{1}{2c}$, the true gap between $\max G_{\mathcal{A}}^-(\epsilon)$ and $\max G_{\mathcal{A}}^+(\epsilon)$ is less than $\frac{1}{2c}$ and the error margin of the binary search estimate the $\mu_{\mathrm{GDP}}$ is also $\frac{1}{2c}$. Therefore, the overall gap is bounded by $\frac{1}{c}$. As for complexity, each binary search has a time complexity of $O(\log(c))$ and the number of binary searches is $2n + 2 = O(\epsilon_h c)$. The overall time complexity of this naive approach is $O(\epsilon_h c \log(c))$. For a complete pseudocode of this naive approach, refer to algorithm 3 in Appendix D.

By leveraging some properties of $\mu_{\mathrm{GDP}}$ and shuffling, the expected number of binary searches needed can be reduced from linear ($2n + 2 \approx c\epsilon_h$) to logarithmic ($O(\log(c\epsilon_h))$). Such reduction will eliminate the logarithmic term in the time complexity from the naive algorithm. The improved algorithm is given as Algorithm 1 below.

---

Algorithm 1: Finding $\mu$ with privacy profiles (optimized).

---

**Input:** $\delta_{\mathcal{A}}$, $\epsilon_h$, $\mu_t$, $c$. (Privacy profile, searching range $\epsilon_h$, reciprocal of error margin)
$n \leftarrow \lceil \sqrt{8c\pi\epsilon_h} \rceil + 1$
$d \leftarrow \frac{\epsilon_h}{n-1}$
$\mu_- \leftarrow 0$
$\mu_+ \leftarrow 0$
$\mathcal{S} = [0, 1, \cdots, n+1]$
Shuffle $\mathcal{S}$
**for** $i = 0$ **to** $n+1$ **do**
    $x^- \leftarrow S[i]d$
    $x^+ \leftarrow (S[i]+1)d$
    **if** $\delta_{\mu^+}(x^-) < \delta_{\mathcal{A}}(x^+)$ **then**
        $\mu^+ \leftarrow \mu_{\text{GDP}}^+(x^-, \delta_{\mathcal{A}}(x^+), \frac{1}{2c}))$
    **end if**
    **if** $\delta_{\mu^-}(x^+) < \delta_{\mathcal{A}}(x^-)$ **then**
        $\mu^- \leftarrow \mu_{\text{GDP}}^-(x^+, \delta_{\mathcal{A}}(x^-), \frac{1}{2c}))$
    **end if**
**end for**
**Output:** $\mu_-$, $\mu_+$ (lower and upper bound of $\mu$).

---

We remark that this algorithm also has better accuracy than the naive algorithm because the lower and upper bounds will be closer while maintaining coverage. Refer to Appendix D for a detailed explanation of this algorithm.

## 4.2 Understanding the tail

With Theorem 4.3, one can verify $(\epsilon_h, \mu)$-head GDP conditions for arbitrarily large $\epsilon_h$ and an arbitrarily precise approximation of $\mu$. While the error in $\mu$ can be quantified by $D$, one gap remains: $\epsilon_h$ can be arbitrarily large but can never truly be $+\infty$. In this subsection, we discuss the gap between $(\epsilon_h, \mu)$-head GDP and actual GDP (which is equivalent to $(+\infty, \mu)$-head GDP). Before giving a solution, we intuitively illustrate the gap between $(\epsilon_h, \mu)$-head GDP and actual GDP. Consider the following two cases:

- GDP with catastrophic failure, where with probability $1-p$, $\mathcal{A}_1$ functions properly as $\mu$-GDP, with probability $p$, $\mathcal{A}_1$ malfunctions and discloses the entire dataset; and

- head-GDP with $\epsilon$-DP, where $\mathcal{A}_2$ is both $(\epsilon_h, \mu)$-head GDP and $(\epsilon_h, 0)$-DP.

The head GDP privacy guarantee lies strictly between those of $\mathcal{A}_1$ and $\mathcal{A}_2$: specifically, $\delta_{\mathcal{A}_1}(\epsilon) < \delta(\epsilon) < \delta_{\mathcal{A}_2}(\epsilon)$. As an interpretation of this inequality, a head GDP privacy guarantee is safer than the original GDP guarantee but with a minuscule probability of failure, and when combined with a very weak $\epsilon$-DP condition, the head GDP will be stronger than the actual GDP. In practice, $\mu$ is rarely above six in GDP, and $\epsilon$ is rarely above 10 in $\epsilon$-DP because more extreme values provide almost no privacy protection [13]. If we verify the head condition up to $\epsilon_h = 100$ (which is not difficult because the time required for verification grows linearly) and take $\mu = 6$, then $p = \delta_\mu(\epsilon_h)$ will be on the order of $10^{-43}$. Also, DP guarantee for $\epsilon$ this large is rarely considered to provide real protection. Hence, we conclude that the gap will not make any notable difference in practice with a proper choice of $\mu$ and $\epsilon_h$.

## 4.3 Amplification

In some cases, one may wish to theoretically mend the gap discussed in the last subsection. This can be achieved by adding extra steps to perturb the output of the algorithm (i.e., via post-processing). We propose the following "clip and rectify" procedure that can turn any $(\epsilon_h, \mu)$-head GDP algorithm into a $\mu$-GDP algorithm at some utility cost.

**Theorem 4.4** *Let $\mathcal{A}$ be an $(\epsilon_h, \mu)$-head GDP algorithm with a numeric output. Assume that $-\infty < y^- < y^+ < +\infty$. Define $\mathcal{C}(y) = \max(\min(y, y^+), y^-)$ and $\mathcal{R}(z) = z + v$, where $v$ is sampled from* $\mathrm{Laplace}(b)$ *with $b = (y^+ - y^-)/\epsilon_h$. Then $\mathcal{R} \circ \mathcal{C} \circ \mathcal{A}$ is $\mu$-GDP.*

Refer to Appendix B.7 for a proof of Theorem 4.4. We remark that, in order to minimize the utility loss, the bounds $y^-$ and $y^+$ should be properly or dynamically chosen and the head condition should be verified to a value of $\epsilon_h$ that is as large as possible.

On the other hand, the performance ($\mu$) of a GDP algorithm may be bottlenecked by the value of its privacy profile near the origin. This problem can be remedied by subsample pre-processing, the impact of which on privacy profiles has been thoroughly examined in [3]. The resulting privacy profile is explicitly given in Theorems 8–10 of [3]. With the help of the GDPT, we can select different subsample ratios and measure $\mu$. For instance, the Laplace example in this paper was originally 1.80-GDP. If we introduce a $50\%$- or $10\%$- Poisson subsampling before the Laplace mechanism, $\mu$ will be reduced to 0.98 or 0.28, respectively. Refer to E.2 for a complete graph of the new GDPTs.

While one could turn to other algorithms or design a new GDP mechanism in unfavourable cases where a candidate algorithm is incompatible with GDP from the start, rectifying these incompatibilities via pre- and post-processing may be more effective and efficient. This is especially true in cases where raw data is not easily accessible. In other cases, the DP mechanism might be inaccessible. This is particularly common for users of proprietary software. While they cannot identify and change the algorithm distributed in binary code, users can still control sensitive information by only approving a subset for release.

## 5 Applications

### 5.1 The Gaussian nature of $\epsilon$-DP and the Laplace mechanism

By our previous analysis of the GDPT, we know that being GDP means that a privacy profile has a quickly vanishing tail (i.e., $\delta(\epsilon)$ must be $O(e^{-\epsilon^2})$). It is remarkable that another single parameter family of DP conditions, the $\epsilon$-DP conditions, is also a property that pertains to the tail of privacy profiles. For any $\epsilon_0$-DP algorithm, the privacy profile must be exactly 0 after $\epsilon_0$. This suggests that $\epsilon$-DP is stronger than GDP. Next, we will quantify this intuition using the tools we developed above.

By Theorem 2.1, we know if $\mathcal{A}$ is $\epsilon_0$-DP, then in the worst case, $\delta_{\mathcal{A}}(\epsilon) = (e^{\epsilon_0} - e^{\epsilon})^+/(1 + e^{\epsilon_0})$.

We consider the GDPT of $\delta_{\mathcal{A}}$, denoted by $G_{\mathcal{A}}$. It is easy to see that, for $\epsilon \geq \epsilon_0$, $G_{\mathcal{A}}(\epsilon) = 0$: we need only consider $\epsilon \in [0, \epsilon_0)$. Let $G_{\delta_{\mathcal{A}}(\epsilon)}$ be denoted by $\mu_\epsilon$. Using the partial derivative of $G_{\mathcal{A}}$ derived in Appendix B.5, we know that $\frac{\partial}{\partial \epsilon} G_{\delta_{\mathcal{A}}(\epsilon)} = \sqrt{2\pi} \exp\left\{\left(\mu_\epsilon^2 + 2\epsilon\right)^2/(8\mu_\epsilon^2)\right\} \left[\Phi(-\frac{\mu_\epsilon^2 + 2\epsilon}{2\mu_\epsilon}) - \Phi(\frac{-\mu_0}{2})\right]$. Then $\mathrm{sign}(\frac{\partial}{\partial \epsilon} G_{\delta_{\mathcal{A}}(\epsilon)}) = \mathrm{sign}(\mu_\epsilon - \mu_0 - 2\epsilon/\mu_0)$. We can conclude that $\mu_\epsilon \leq \mu_0$ and, further, that $G_{\mathcal{A}}(\epsilon)$ is strictly decreasing on $[0, \epsilon_0)$. By Theorem 4.1, we know that $\mathcal{A}$ is $\mu_0$-GDP. This finding can be more generally formulated as the following theorem.

**Theorem 5.1** *Any $(\epsilon, 0)$-DP algorithm is also $\mu$-GDP for $\mu = -2\Phi^{-1}(1/(1 + e^{\epsilon})) \leq \sqrt{\pi/2}\epsilon$.*

[13] pointed out that the DP guarantees of the Laplace mechanism are stronger than those correspondingly provided by $\epsilon$-DP. We reaffirm this difference by showing that it still exists under the GDP framework. The Laplace mechanism satisfies $\mu$-GDP for $\mu$ smaller than the bound given in Theorem 5.1. The GDPTs presented in Appendix E.1 illustrate this difference.

### 5.2 Handling composition with GDP

In practice, it is rare for a dataset to go through DP algorithms only once. Multiple statistics may be of interest or one statistic may require multiple inquiries to acquire. DP algorithms applied to the same dataset multiple times are usually still DP but with worse privacy parameters. Composition theorems quantitatively trace privacy loss and provide a privacy parameter for the ensemble. However, not only is exact composition an intrinsically (#P-)hard problem [29], but the conclusions of composition theorems are also often problematic. Take traditional $(\epsilon, \delta)$-DP as an example. [24] gives an optimal composition theorem, but the composition of two $(\epsilon, \delta)$-DP algorithms cannot be characterized under the $(\epsilon, \delta)$-DP framework. This result damages interpretability because the representation of

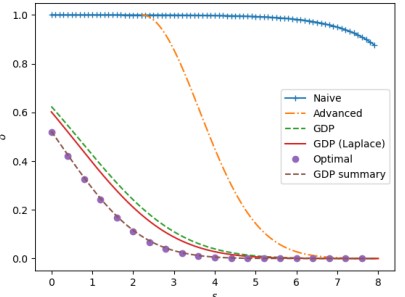

Figure 3: The plot of privacy guarantee under different methods.

Table 1: Minimum values of $\epsilon$ to achieve corresponding $(\epsilon, \delta)$-DP.

| Method | $\delta$   $10^{-1}$ | $10^{-2}$ | $10^{-3}$ | $10^{-4}$ |
|---|---|---|---|---|
| Basic | 9.89 | 9.99 | 10 | 10 |
| Advanced | 5.25 | 6.51 | 7.47 | 8.28 |
| RDP | 12.14 | 17.17 | 21.03 | 24.28 |
| GDP | 3.1 | 5.06 | 6.47 | 7.62 |
| GDP (Lap) | 2.87 | 4.74 | 6.09 | 7.19 |
| Optimal | 2.12 | 3.64 | 4.76 | 5.28 |
| GDP summary | 2.14 | 3.73 | 4.87 | 5.80 |

a composition will no longer be in two parameters. This type of flaw is the major motivation for a GDP characterization of algorithms derived under other DP frameworks. The composition of GDP algorithms is easy, exact, and closed: the composition of a $\mu_1$- and $\mu_2$-GDP algorithm is simply $\sqrt{\mu_1^2 + \mu_2^2}$-GDP. GDP also has a special central limit theorem which implies that, for all privacy definitions that retain hypothesis testing with proper scaling, the privacy guarantee of a composition converges to GDP in the limit. In this subsection, we demonstrate that GDP is a powerful tool for composition by unifying other notions under the GDP framework and then using the GDP composition theorem. As baselines, we select basic composition [14], advanced composition [15] and Rényi-DP [28].

We consider the 50-fold composition of $0.2$-DP algorithms. In this setting, the basic composition is pessimistic and says that the composition will be 10-DP, which means there is next to no privacy guarantee. According to corollary 1 of [28], the bound given by RDP is even looser. Refer to Figure 3 for the results of other theorems.

We next consider composition using the proposed measurement method. According to Theorem 5.1, a $0.2$-DP algorithm is $0.2505$-GDP. If the algorithm is the Laplace mechanism, then the algorithm in Appendix D can tighten $\mu$ to $0.2391$. To compute $\mu$ for a 50-fold composition, we simply multiply the original $\mu$ by $\sqrt{50}$. The result is $1.771$-GDP ($1.691$ for the Laplace mechanism). In this case, distinguishing two neighbouring datasets is as hard as distinguishing between $N(0, 1)$ and $N(1.771, 1)$ on the basis of a single observation.

In this particular case, the ground truth can be derived from the optimal composition theorem [24]. We present the results from the optimal composition theorem in Table 1 and Figure 3 for comparison, but we do not consider the optimal composition theorem to be generally superior because the ground truth is not easy to compute and because the former method is not as interpretable and only works for algorithms whose DP guarantees are fixed at $(\epsilon, \delta)$. However, by applying the GDPT, the privacy guarantee of the optimal composition theorem can be summarised as $1.420$-GDP . Compared to the central limit theorem in [13] which yields $\mu = \sqrt{2}$ (with an unknown asymptotic approximation error) in the same setting, the tractable numerical procedure of GDPT provides a satisfying result.

## 6 Conclusion and Future Work

In this paper, we provided both an analytic perspective of and engineering tools for the GDP framework. By using the new notions we proposed, we devised solutions to three aspects of GDP: identification, amplification, and measurement. The developments in this paper suggest numerous interesting directions for future work. First, the more refined methods can be derived to expand the toolbox of rectification for more versatility. Second, the measurement procedure can be combined with the rectification procedure. Incrementally introducing more pre- and post-processing steps and dynamically checking whether privacy guarantees are already satisfactory can also be explored. Lastly, the idea underlying the GDPT can be generalized to other parameterized DP notions like CDP or RDP to enrich tractability and visualizability in the DP literature.

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
