# A   Appendix

# B   Appendix: Proofs

**Proof B.1** *Proof of Theorem 2.1:*

*Sufficiency:*

*When $\epsilon \geq \epsilon_0$, the sufficiency is trivial as $\delta = \delta_0$.*

*When $\epsilon < \epsilon_0$, given that $\mathcal{A}$ is $(\epsilon_0, \delta_0)$-DP, by the definition, for any pair of datasets $S$ and $S'$ that differ in the record of a single individual and any event $E$,*

$$P[\mathcal{A}(S) \in E] \leq e^{\epsilon_0} P\left[\mathcal{A}(S') \in E\right] + \delta_0.$$

*When $P\left[\mathcal{A}(S') \in E\right] \leq \frac{1-\delta_0}{1+e^{\epsilon_0}} := c_0$,*

$$
\begin{aligned}
P[\mathcal{A}(S) \in E] &\leq e^{\epsilon_0} P\left[\mathcal{A}(S') \in E\right] + \delta_0 \\
&\leq (e^{\epsilon_0} + e^{\epsilon} - e^{\epsilon}) P\left[\mathcal{A}(S') \in E\right] + \delta_0 + \delta - \delta \\
&\leq e^{\epsilon_0} P\left[\mathcal{A}(S') \in E\right] + \delta + (e^{\epsilon_0} - e^{\epsilon})c_0 + \delta_0 - \delta \\
&\leq e^{\epsilon} P\left[\mathcal{A}(S') \in E\right] + \delta + (e^{\epsilon_0} - e^{\epsilon})c_0 - \frac{(1-\delta_0)(e^{\epsilon_0} - e^{\epsilon})}{1 + e^{\epsilon_0}} \\
&\leq e^{\epsilon} P\left[\mathcal{A}(S') \in E\right] + \delta.
\end{aligned}
$$

*When $c_0 \leq P\left[\mathcal{A}(S') \in E\right] \leq 1$,*

$$
\begin{aligned}
P[\mathcal{A}(S) \in E] &= 1 - P[\mathcal{A}(S) \in E^c] \\
&\leq 1 - e^{-\epsilon_0}(P\left[\mathcal{A}(S') \in E^c\right] - \delta_0) \\
&= 1 - e^{-\epsilon_0}(1 - P\left[\mathcal{A}(S') \in E\right] - \delta_0) \\
&= 1 - e^{-\epsilon_0} + e^{-\epsilon_0} P\left[\mathcal{A}(S') \in E\right] + e^{-\epsilon_0}\delta_0 \\
&= 1 - e^{-\epsilon_0} + e^{-\epsilon_0}\delta_0 + \delta - \delta + (e^{-\epsilon_0} + e^{\epsilon} - e^{\epsilon}) P\left[\mathcal{A}(S') \in E\right] \\
&= e^{\epsilon} P\left[\mathcal{A}(S') \in E\right] + \delta + 1 - e^{-\epsilon_0} + e^{-\epsilon_0}\delta_0 - \delta + (e^{-\epsilon_0} - e^{\epsilon}) P\left[\mathcal{A}(S') \in E\right] \\
&\leq e^{\epsilon} P\left[\mathcal{A}(S') \in E\right] + \delta + 1 - e^{-\epsilon_0} + e^{-\epsilon_0}\delta_0 - \delta + (e^{-\epsilon_0} - e^{\epsilon})c_0 \\
&= e^{\epsilon} P\left[\mathcal{A}(S') \in E\right] + \delta + (1 - \delta_0)\left(\frac{e^{-\epsilon_0} - e^{\epsilon}}{1 + e^{\epsilon_0}} - e^{-\epsilon_0}\right) + 1 - \delta \\
&\leq e^{\epsilon} P\left[\mathcal{A}(S') \in E\right] + \delta + (1 - \delta_0)\left(\frac{e^{-\epsilon_0} - e^{\epsilon}}{1 + e^{\epsilon_0}} - e^{-\epsilon_0} + 1 + \frac{e^{\epsilon} - e^{\epsilon_0}}{1 + e^{\epsilon_0}}\right) \\
&= e^{\epsilon} P\left[\mathcal{A}(S') \in E\right] + \delta.
\end{aligned}
$$

*Necessity:*

*We prove the necessity by giving a specific $(\epsilon_0, \delta_0)$-DP algorithm $\mathcal{A}$ such that $\delta_{\mathcal{A}}(\epsilon)$ is exactly $\delta_0 + \frac{(1-\delta_0)(e^{\epsilon_0} - e^{\epsilon})^+}{1+e^{\epsilon_0}}$.*

*Define $\Omega_e = \{1, 2, 3, 4\}$ and $\Omega_S = \{0, 1\}$. Let $\epsilon \geq 0$, $0 \leq \delta_0 \leq 1$ and denote $\frac{e^{\epsilon_0}}{1+e^{\epsilon_0}}$ as $\alpha_0$. Let $\mathcal{A}$ be a randomized algorithm that take a single point from $\Omega_S$ and generate output as follows:*

$$
\begin{cases}
P(\mathcal{A}(S) = 1 \mid S = 0) = \delta_0, \\
P(\mathcal{A}(S) = 2 \mid S = 0) = 0, \\
P(\mathcal{A}(S) = 3 \mid S = 0) = (1 - \delta_0)\alpha_0, \\
P(\mathcal{A}(S) = 4 \mid S = 0) = (1 - \delta_0)(1 - \alpha_0),
\end{cases}
\qquad
\begin{cases}
P(\mathcal{A}(S) = 1 \mid S = 1) = 0, \\
P(\mathcal{A}(S) = 2 \mid S = 1) = \delta_0, \\
P(\mathcal{A}(S) = 3 \mid S = 1) = (1 - \delta_0)(1 - \alpha_0), \\
P(\mathcal{A}(S) = 4 \mid S = 1) = (1 - \delta_0)\alpha_0.
\end{cases}
$$

*By definition, $\delta(\epsilon)$ is the smallest $\delta$ such that $P(\mathcal{A}(S) \subset E \mid S = s) \leq e^{\epsilon} P(\mathcal{A}(S) \subset E \mid S = 1 - s) + \delta$ holds true for all $E \subset \Omega_e$ and $s \in \Omega_S$. By checking all 64 combinations, we can conclude that $\delta_{\mathcal{A}}(\epsilon) = \delta_0 + \frac{(1-\delta_0)(e^{\epsilon_0} - e^{\epsilon})^+}{1+e^{\epsilon_0}}$.*

**Proof B.2** *Proof of Lemma 3:*

*It is well known that [2], for $t < 0$:*

$$\frac{1}{-t + \sqrt{t^2 + 4}} < \sqrt{\frac{\pi}{2}} \exp\left(\frac{t^2}{2}\right) \Phi(t) < \frac{1}{-t + \sqrt{t^2 + \frac{8}{\pi}}}.$$

*Let $a = \left(-\frac{\varepsilon}{\mu} + \frac{\mu}{2}\right)$ and $b = \left(-\frac{\varepsilon}{\mu} - \frac{\mu}{2}\right)$,*

$$\overline{\lim_{\epsilon \to \infty}} \, \delta_\mu(\epsilon) = \overline{\lim_{\epsilon \to \infty}} \, \Phi(a) - e^\epsilon \Phi(b)$$

$$\leq \sqrt{\frac{2}{\pi}} \overline{\lim_{\epsilon \to \infty}} \, \frac{\exp\left(\frac{-a^2}{2}\right)}{-a + \sqrt{a^2 + \frac{8}{\pi}}} - \frac{\exp\left(\frac{-b^2}{2} + \epsilon\right)}{-b + \sqrt{b^2 + 4}}.$$

$$= \sqrt{\frac{2}{\pi}} \overline{\lim_{\epsilon \to \infty}} \, \exp\left(\frac{-a^2}{2}\right) \left(\frac{1}{-a + \sqrt{a^2 + \frac{8}{\pi}}} - \frac{1}{-b + \sqrt{b^2 + 4}}\right).$$

$$\leq \sqrt{\frac{2}{\pi}} \overline{\lim_{\epsilon \to \infty}} \, \exp\left(\frac{-a^2}{2}\right) \left(\frac{-1}{a}\right).$$

$$= 0.$$

$$\underline{\lim_{\epsilon \to \infty}} \, \delta_\mu(\epsilon) = \underline{\lim_{\epsilon \to \infty}} \, \Phi(a) - e^\epsilon \Phi(b)$$

$$\geq \sqrt{\frac{2}{\pi}} \underline{\lim_{\epsilon \to \infty}} \, \frac{\exp\left(\frac{-a^2}{2}\right)}{-a + \sqrt{a^2 + 4}} - \frac{\exp\left(\frac{-b^2}{2} + \epsilon\right)}{-b + \sqrt{b^2 + \frac{8}{\pi}}}.$$

$$= \sqrt{\frac{2}{\pi}} \underline{\lim_{\epsilon \to \infty}} \, \exp\left(\frac{-a^2}{2}\right) \left(\frac{1}{-a + \sqrt{a^2 + 4}} - \frac{1}{-b + \sqrt{b^2 + \frac{8}{\pi}}}\right).$$

$$\geq \sqrt{\frac{2}{\pi}} \underline{\lim_{\epsilon \to \infty}} \, \exp\left(\frac{-a^2}{2}\right) \left(\frac{-1}{b}\right).$$

$$= 0.$$

*Therefore,*

$$\lim_{\epsilon \to \infty} \delta_\mu(\epsilon) = 0. \tag{3}$$

*It is easy to see that,*

$$\lim_{\epsilon \to \infty} \tilde{\delta}_\mu(\epsilon) = \lim_{\epsilon \to \infty} \frac{\mu e^{-a^2/2}}{\sqrt{2\pi a^2}} = 0 \tag{4}$$

*By L'Hospital's rule:*

$$\lim_{\epsilon \to \infty} \frac{\tilde{\delta}_\mu(\epsilon)}{\delta_\mu(\epsilon)} = \lim_{\epsilon \to \infty} \frac{\tilde{\delta}'_\mu(\epsilon)}{\delta'_\mu(\epsilon)}$$

$$= \lim_{\epsilon \to \infty} -\frac{e^{-\frac{a^2}{2}} \left(a^2 + 2\right)}{\sqrt{2\pi a^3}} \bigg/ e^\epsilon \Phi(b)$$

$$= \lim_{\epsilon \to \infty} \frac{e^{-\frac{b^2}{2}} \Phi(b)}{\sqrt{2\pi b}}$$

$$= \lim_{b \to -\infty} \frac{e^{-\frac{b^2}{2}} \Phi(b)}{\sqrt{2\pi b}}$$

$$= 1.$$

**Proof B.3** *Proof of Theorem 3.2:*

*Sufficiency:*

*If $\mathcal{A}$ is $\mu$-GDP. Then $\varlimsup\limits_{\epsilon\to+\infty} G_{\mathcal{A}}(\epsilon) \leq \varlimsup\limits_{\epsilon\to+\infty} G_{\delta_\mu}(\epsilon) = \mu$.*

*Necessity:*

*If $\varlimsup\limits_{\epsilon\to+\infty} G_{\mathcal{A}}(\epsilon) = \mu < +\infty$, there must be a $\epsilon_t > 0$ such that $\mathcal{A}$ is $(\epsilon_t, \mu_0 + 1)$-tail GDP.*

*Notice that $\lim\limits_{\mu\to\infty} \delta_\mu(\epsilon_t) = 1$, we can pick $\mu_1 > \mu_0$ large enough such that $\delta_{\mu_1}(\epsilon_t) > \delta_{\mathcal{A}}(0)$.*

*This is possible because by Theorem 2.1, $\delta_{\mathcal{A}}(0) < 1$. Then for $\epsilon \in [0, \epsilon_t)$, $\delta_{\mathcal{A}}(\epsilon) \leq \delta_{\mathcal{A}}(0) \leq \delta_{\mu_1}(\epsilon_t) \leq \delta_{\mu_1}(\epsilon)$. $\mathcal{A}$ is both $(\epsilon_t, \mu)$-head and tail GDP for $\mu = \mu_0 + \mu_1 + 1$. $\mathcal{A}$ is GDP as desired.*

**Proof B.4** *Proof of Theorem 3.3:*

*Let $\varlimsup\limits_{\epsilon\to+\infty} G_f(\epsilon) = \mu_t$.*

*First we show that $\varlimsup\limits_{\epsilon\to\infty} \frac{\epsilon^2}{-2\log\delta_{\mathcal{A}}(\epsilon)} \leq \mu_t^2$:*

*By the definition the limit, for any $\mu_0 > \mu_t$, for sufficient large $\epsilon$, $G_f(\epsilon) < \mu_0$ and further $\delta_{\mathcal{A}}(\epsilon) \leq \delta_{\mu_0}(\epsilon)$. Hence, $\varlimsup\limits_{\epsilon\to\infty} \frac{\delta_{\mathcal{A}}(\epsilon)}{\delta_{\mu_0}(\epsilon)} \leq 1$. By Lemma 3, $\varlimsup\limits_{\epsilon\to\infty} \frac{\delta_{\mathcal{A}}(\epsilon)}{\tilde{\delta}_{\mu_0}(\epsilon)} \leq 1$.*

*Then $\lim\limits_{\epsilon\to\infty} \frac{\epsilon^2}{-2\log\delta_{\mathcal{A}}(\epsilon)} \leq \lim\limits_{\epsilon\to\infty} \frac{\epsilon^2}{-2\log\tilde{\delta}_{\mu_0}(\epsilon)} = \mu_0^2$.*

*$\lim\limits_{\epsilon\to\infty} \frac{\epsilon^2}{-2\log\delta_{\mathcal{A}}(\epsilon)} \leq \mu_t$ as desired as we take $\mu_0 \to \mu_t$.*

*Next we show that $\varlimsup\limits_{\epsilon\to\infty} \frac{\epsilon^2}{-2\log\delta_{\mathcal{A}}(\epsilon)} \geq \mu_t^2$:*

*If $\varlimsup\limits_{\epsilon\to\infty} \frac{\epsilon^2}{-2\log\delta_{\mathcal{A}}(\epsilon)} = \mu_0^2 < \mu_t^2$, then by Lemma 3,*

$$\varlimsup\limits_{\epsilon\to\infty} \frac{\epsilon^2}{-2\log\delta_{\mathcal{A}}(\epsilon)} - \frac{\epsilon^2}{-2\log\delta_{\mu_t}(\epsilon)} = \varlimsup\limits_{\epsilon\to\infty} \frac{\epsilon^2}{-2\log\delta_{\mathcal{A}}(\epsilon)} - \varlimsup\limits_{\epsilon\to\infty} \frac{\epsilon^2}{-2\log\tilde{\delta}_{\mu_t}(\epsilon)}$$
$$< \mu_0^2 - \mu_t^2$$

*Then for a sufficiently large $\epsilon_0$,*

$$\frac{\epsilon_0^2}{-2\log\delta_{\mathcal{A}}(\epsilon_0)} - \frac{\epsilon_0^2}{-2\log\delta_{\mu_0}(\epsilon_0)} < 0.$$

*Since $\log$ is an increasing function, it follows that $\delta_{\mathcal{A}}(\epsilon_0) < \delta_{\mu_0}(\epsilon_0)$. Then $\varlimsup\limits_{\epsilon\to+\infty} G_f(\epsilon) \leq \mu_0 < \mu_t$, which is a contradiction.*

**Proof B.5** *Proof of Theorem 4.2:*

*Let $G_\mu(\epsilon) = F(\epsilon, \delta_\mu(\epsilon))$ and $F(x, y) = \mu_{\text{GDP}}(x, y)$.*

*By definition of $\mu_{\text{GDP}}$, $G_\mu(\epsilon) = \mu$.*

*On one hand,* $\begin{cases} \dfrac{\partial G_\mu(\epsilon)}{\partial\epsilon} = \dfrac{\partial\mu}{\partial\epsilon} = 0, \\ \dfrac{\partial G_\mu(\epsilon)}{\partial\mu} = \dfrac{\partial\mu}{\partial\mu} = 1. \end{cases}$

*On the other hand, by chain rule,* $\begin{cases} \dfrac{\partial G_\mu(\epsilon)}{\partial\epsilon} = \dfrac{\partial F}{\partial x} + \dfrac{\partial F}{\partial y}\dfrac{\partial\delta_\mu(\epsilon)}{\partial\epsilon}, \\ \dfrac{\partial G_\mu(\epsilon)}{\partial\mu} = \dfrac{\partial F}{\partial y}\dfrac{\partial\delta_\mu(\epsilon)}{\partial\mu}. \end{cases}$

*Therefore,*
$$\begin{cases} \dfrac{\partial F}{\partial y} = (\dfrac{\partial \delta_\mu(\epsilon)}{\partial \mu})^{-1}, \\ \dfrac{\partial F}{\partial x} = -(\dfrac{\partial \delta_\mu(\epsilon)}{\partial \mu})^{-1}\dfrac{\partial \delta_\mu(\epsilon)}{\partial \epsilon}. \end{cases}$$

*Using the close forms, $\frac{\partial \delta_\mu(\epsilon)}{\partial \epsilon}$ and $\frac{\partial \delta_\mu(\epsilon)}{\partial \mu}$ can be directly computed:*

$$\begin{cases} \dfrac{\partial \delta_\mu(\epsilon)}{\partial \epsilon} = -e^\epsilon \Phi(-\dfrac{\mu^2 + 2\epsilon}{2\mu}), \\ \dfrac{\partial \delta_\mu(\epsilon)}{\partial \mu} = \dfrac{e^{-\frac{(\mu^2-2\epsilon)^2}{8\mu^2}}}{\sqrt{2\pi}}. \end{cases}$$

*Hence,*
$$\begin{cases} \dfrac{\partial F}{\partial x} = \sqrt{2\pi}e^{\frac{(\mu^2+2\epsilon)^2}{8\mu^2}}\Phi(-\dfrac{\mu^2+2\epsilon}{2\mu}) \le \sqrt{2\pi}e^{\frac{\mu^2}{8}}\Phi(-\dfrac{\mu}{2}) \le \dfrac{\sqrt{2\pi}}{2}, \\ \dfrac{\partial F}{\partial y} = \sqrt{2\pi}e^{\frac{(\mu^2-2\epsilon)^2}{8\mu^2}} > 0. \end{cases}$$

*Notice that $\frac{\partial F}{\partial x} = \sqrt{2\pi}e^{\frac{(\mu^2+2\epsilon)^2}{8\mu^2}}\Phi(-\frac{\mu^2+2\epsilon}{2\mu}) > 0$, combined with the fact that $\frac{\partial F}{\partial x} \le \frac{\sqrt{2\pi}}{2}$, we can conclude that $0 \le \frac{\partial \mu_{GDP}(\epsilon,\delta)}{\partial \epsilon} \le \frac{\sqrt{2\pi}}{2}$. By $\frac{\partial F}{\partial y} > 0$, we can see GDPT is order preserving.*

**Proof B.6** *Proof of Theorem 4.3:*

*We now consider the gap between $\max_{i\in\{0,\cdots,n\}}\{G_\mathcal{A}^-(x_i)\}$ and $\max_{i\in\{0,\cdots,n+1\}}\{G_\mathcal{A}^+(x_i)\}$ bound the length of $[\mu^-,\mu^+]$ in two cases.*

*Case 1: If $\max_{i\in\{0,\cdots,n+1\}}\{G_\mathcal{A}^+(x_i)\} = G_\mathcal{A}^+(x_0)$, then $\max_{i\in\{0,\cdots,n+1\}}\{G_\mathcal{A}^+(x_i)\} = G_\mathcal{A}^+(x_0) = \mu_{GDP}(D, \delta_\mathcal{A}(0)) \le \mu_{GDP}(0, \delta_\mathcal{A}(0)) + \frac{\sqrt{2\pi}D}{2}$. Therefore,*

$$\max_{\epsilon\in[0,\epsilon_h]} G(\epsilon) \le G_\mathcal{A}^+(x_0) \le \{G_\mathcal{A}^-(x_0)\} + \frac{\sqrt{2\pi}D}{2}.$$

*Case 2: If $\max_{i\in\{0,\cdots,n+1\}}\{G_\mathcal{A}^+(x_i)\} \ne G_\mathcal{A}^+(x_0)$, then by the order preserving property, the optimal $\mu$ lies in $[\mu^-, \mu^+]$, where $\mu^- = \max(\mu_h, \max_{i\in\{0,\cdots,n\}}\{G_\mathcal{A}^-(x_i)\})$ and $\mu^+ = \max(\mu_h, \max_{i\in\{1,\cdots,n+1\}}\{G_\mathcal{A}^+(x_i)\})$. Notice that*

$$\max_{i\in\{0,\cdots,n\}}\{G_\mathcal{A}^-(x_i)\} = \max_{i\in\{0,\cdots,n\}}\{\mu_{GDP}(x_i, \delta_\mathcal{A}(x_{i+1}))\} = \max_{i\in\{1,\cdots,n+1\}}\{\mu_{GDP}(x_{i-1}, \delta_\mathcal{A}(x_i))\}$$
$$\ge \max_{i\in\{1,\cdots,n+1\}}\{\mu_{GDP}(x_{i+1}, \delta_\mathcal{A}(x_i)) - \sqrt{2\pi}D\}$$
$$\ge \max_{i\in\{1,\cdots,n+1\}}\{G_\mathcal{A}^+(x_i)\} - \sqrt{2\pi}D.$$

*In both cases the gap is no greater than $\sqrt{2\pi}D$ as desired.*

**Proof B.7** *Proof of Theorem 4.4:*

*By the definition of $\mathcal{C}$, $\mathcal{C} \circ \mathcal{A}$ is bounded in $[y^-, y+]$. Therefore the global sensitivity of $\mathcal{C} \circ \mathcal{A}$ is no greater than $y^+ - y^-$. Then $\mathcal{R} \circ \mathcal{C} \circ \mathcal{A}$ is a special case of the Laplace mechanism. By [3], $\mathcal{R} \circ \mathcal{C} \circ \mathcal{A}$ is $\epsilon_h$-DP. Then $\delta_{\mathcal{R}\circ\mathcal{C}\circ\mathcal{A}}(\epsilon) = 0 < \delta_\mu(\epsilon)$ for any $\epsilon \ge \epsilon_h$.*

*In addition, because of the post-processing property, $\delta_{\mathcal{R}\circ\mathcal{C}\circ\mathcal{A}}(\epsilon) \le \delta_\mathcal{A}(\epsilon) < \delta_\mu(\epsilon)$ for any $\epsilon < \epsilon_h$.*

*Therefore, $\mathcal{R} \circ \mathcal{C} \circ \mathcal{A}$ is $\mu$-GDP.*

## C  Appendix: Refining the privacy profile

Given a trade-off function $\sigma = f(\epsilon, \delta)$ and a fixed parameter $\sigma$. From definition of the trade-off function it is instant that the for any $(\epsilon, \delta) \in \Omega = \{(\epsilon, \delta) \mid \sigma = f(\epsilon, \delta)\}$, $(\epsilon, \delta)$-DP is guaranteed.

Then, $(\epsilon, \delta)$-DP is also guaranteed if there is a $(\epsilon_0, \delta_0) \in \Omega$ such that $(\epsilon_0, \delta_0)$-DP implies $(\epsilon, \delta)$-DP. Therefore,

$$\delta_{\mathcal{A}}(\epsilon) = \min\left(\{\delta \mid \sigma = f(\epsilon_0, \delta_0) \text{ and } \delta \geq \delta_0 + \frac{(1 - \delta_0)(e^{\epsilon_0} - e^\epsilon)^+}{1 + e^{\epsilon_0}}\}\right).$$

Notice that by theorem 2.1, $(\epsilon_0, \delta_0)$-DP implies $(\epsilon, \delta)$ with $\delta < \delta_0$ only if $\epsilon < \epsilon_0$, we rewrite the $\delta_{\mathcal{A}}(\epsilon)$ as:

$$\delta_{\mathcal{A}}(\epsilon) = \inf_{\epsilon_0 \in [\epsilon, \infty)} g(\epsilon, \epsilon_0),$$

where $g(\epsilon, \epsilon_0) := (1 - \hat{\delta_{\mathcal{A}}}(\epsilon_0)) \frac{e^{\epsilon_0} - e^\epsilon}{e^{\epsilon_0} + 1} + \hat{\delta_{\mathcal{A}}}(\epsilon_0)$ and $\hat{\delta_{\mathcal{A}}}$ is the naive privacy profile defined implicitly by $\sigma = f(\epsilon_0, \delta_0)$. For continuously differentiable $f$, the minimum value of the right-hand side can be found be take the derivative:

$$\frac{\partial g(\epsilon, \epsilon_0)}{\partial \epsilon_0} = \frac{1 + e^\epsilon}{(1 + e^{\epsilon_0})^2} \left[ \hat{\delta_{\mathcal{A}}}'(\epsilon_0) + e^{\epsilon_0}(1 - \hat{\delta_{\mathcal{A}}}(\epsilon_0) + \hat{\delta_{\mathcal{A}}}'(\epsilon_0)) \right].$$

We remark that the sign of $\frac{\partial g(\epsilon, \epsilon_0)}{\partial \epsilon_0}$ does not depend on $\epsilon$ when $\epsilon > \epsilon_0$. For both of our example 2 and 3, we both find a particular value $\epsilon^i$ such that $Sign(\frac{\partial g(\epsilon, \epsilon_0)}{\partial \epsilon_0}) = -Sign(\epsilon - \epsilon^i)$. This means for $\epsilon \geq \epsilon^i$, $\delta_{\mathcal{A}}(\epsilon) = \hat{\delta_{\mathcal{A}}}(\epsilon)$ and otherwise $\delta_{\mathcal{A}}(\epsilon)$ equals to the $\delta$ value derived from $(\epsilon^i, \hat{\delta_{\mathcal{A}}}(\epsilon^i))$.

There is an interesting byproduct or the privacy profile refinement. Theoretically, the privacy profile refinement can also be used to improve an algorithm's utility. For example, the projected noisy SGD algorithm in [17] is $(\epsilon, \delta)$-DP and the trade-off function is $\sigma = -C \log(\delta_0)/\epsilon_0$. To achieve $(0.2, e^{-2})$-DP, it appears that $\sigma$ needs to be chosen as $-C \log(e^{-2})/0.2 = 10C$. $(\epsilon, \delta)$-DP implies $(0.2, e^{-2})$-DP when $\delta + (1 - \delta)(e^\epsilon - e^{0.2})^+/(1 + e^\epsilon) = e^{-2}$. Numerical methods suggest that, by choosing $\epsilon \approx 0.334$ and $\delta \approx 0.067$, $(\epsilon, \delta)$-DP implies $(0.2, e^{-2})$-DP but $\sigma = -C \log(\delta)/\epsilon \approx 8.086C < 10C$. Therefore, the desired level of DP can be achieved with a lower noise parameter. However, this type of refinement majorly affects privacy profile around the origin and therefore minor in practice.

## D    Behind efficient head measurement algorithm

First we formalize the binary search algorithm to find $\mu_{\text{GDP}}$:

---

### Algorithm 2: Binary search

**Input:** $\epsilon$, $\delta$, $b$. (The $(\epsilon, \delta)$-pair, searching range, error margin)
$\mu_- \leftarrow 0$
$\mu_+ \leftarrow \mu_{\max}$
**repeat**
    $\mu = \frac{\mu^+ + \mu^-}{2}$
    **if** $\delta_\mu(\epsilon) > \delta$ **then**
        $\mu^+ \leftarrow \mu$
    **else**
        $\mu^- \leftarrow \mu$
    **end if**
**until** $\mu^+ - \mu^- < b$
**Output:** $\mu_-$, $\mu_+$ (lower and upper bound of $\mu$).

---

It is possible to drop the need for the searching range $\mu_{\max}$ for this algorithm (e.g., exponentially search for an upper bound first or conduct a binary search on $\arctan \mu$ instead). We keep this input for clarity and simplicity. $\mu_{\max}$ can be set to a large constant for convenience, for example, 10. If the outputted $\mu^+$ equals the preset value (10), the privacy profile fails to imply 10-GDP. In practice, GDP with $\mu \geq 6$ already provides almost no privacy protection [13].

With the formal definition of binary search, an exhaustive iteration method to bound the staircase functions outlined in Theorem 4.3 can be formally written as follows:

---

Algorithm 3: Finding $\mu$ with privacy profiles (naive).

---

**Input:** $\delta_{\mathcal{A}}, \epsilon_h, c$. (Privacy profile, searching range $\epsilon_h$, reciprocal of error margin)
$n \leftarrow \lceil \sqrt{8c\pi\epsilon_h} \rceil + 1$
$d \leftarrow \frac{\epsilon_h}{n-1}$
$\mu_- \leftarrow 0$
$\mu_+ \leftarrow 0$
**for** $i = 0$ **to** $n + 1$ **do**
    $x^- \leftarrow id$
    $x^+ \leftarrow (i + 1)d$
    $\mu_+ \leftarrow \max(\mu_+, \mu_{\text{GDP}}^+(x^-, \delta_{\mathcal{A}}(x^+), \frac{1}{2c}))$
    $\mu_- \leftarrow \max(\mu_-, \mu_{\text{GDP}}^-(x^+, \delta_{\mathcal{A}}(x^-), \frac{1}{2c}))$
    $i \leftarrow i + 1$
**end for**
**Output:** $\mu^+, \mu^-$.

---

To transform this naive algorithm into the optimized one. The first key observation is that the reassignment of $\mu_+$ and $\mu_-$ can be optimized.

We take $\mu_+ \leftarrow \max(\mu_+, \mu_{\text{GDP}}^+(x^-, \delta_{\mathcal{A}}(x^+), \frac{1}{2c}))$ for example, same optimization can be applied to $\mu_- \leftarrow \max(\mu_-, \mu_{\text{GDP}}^-(x^+, \delta_{\mathcal{A}}(x^-), \frac{1}{2c})))$ as well. The naive operation, $\mu_+ \leftarrow \max(\mu_+, \mu_{\text{GDP}}^+(x^-, \delta_{\mathcal{A}}(x^+), \frac{1}{2c}))$ can be optimized into "If $\delta_{\mu^+}(x^-) < \delta_{\mathcal{A}}(x^+)$, then $\mu^+ \leftarrow \mu_{\text{GDP}}^+(x^-, \delta_{\mathcal{A}}(x^+), \frac{1}{2c}))$" without lost of accuracy. To see this, we list all three possibilities as follows:

- Case 1: $\mu^+ < \mu_{\text{GDP}}(x^-, \delta_{\mathcal{A}}(x^+)) \leq \mu_{\text{GDP}}^+(x^-, \delta_{\mathcal{A}}(x^+), \frac{1}{2c}))$.

- Case 2: $\mu_{\text{GDP}}(x^-, \delta_{\mathcal{A}}(x^+)) \leq \mu^+ \leq \mu_{\text{GDP}}^+(x^-, \delta_{\mathcal{A}}(x^+), \frac{1}{2c}))$.

- Case 3: $\mu_{\text{GDP}}(x^-, \delta_{\mathcal{A}}(x^+)) \leq \mu_{\text{GDP}}^+(x^-, \delta_{\mathcal{A}}(x^+), \frac{1}{2c})) < \mu^+$.

In case 1, both of the naive operation and the optimized operation will update $\mu^+$ to $\mu_{\text{GDP}}^+(x^-, \delta_{\mathcal{A}}(x^+), \frac{1}{2c}))$.

In case 2, the optimized operation will do nothing, because the test $\delta_{\mu^+}(x^-) < \delta_{\mathcal{A}}(x^+)$ will fail. The naive operation will update $\mu^+$ due to the error of binary search, which should be avoided.

In case 3, the optimized operation will do nothing, because the test $\delta_{\mu^+}(x^-) < \delta_{\mathcal{A}}(x^+)$ will fail. The naive operation will also do nothing because the max operator will choose $\mu^+$.

To sum up, the optimized operation always give a more accurate update.

The second insight is that we want to avoid case 1 because only in case 1 a binary search is needed. Notice that case 1 happens only if $\delta_{\mu^+}(x^-) < \delta_{\mathcal{A}}(x^+)$, which is equivalent to $\mu^+ < \mu_{\text{GDP}}(x^-, \delta_{\mathcal{A}}(x^+))$. In the $k + 1$ round of loop, the condition $\mu^+ < \mu_{\text{GDP}}(x^-, \delta_{\mathcal{A}}(x^+))$ holds true only if for all $j \in \{0, \cdots, k\}$, $\mu_{\text{GDP}}(x_j^-, \delta_{\mathcal{A}}(x_j^+)) < \mu_{\text{GDP}}(x^-, \delta_{\mathcal{A}}(x^+))$, where $x_j^-$ and $x_j^+$ are the values of $x^-$ and $x^+$ in the round $j$. This inspire us to shuffle $x_i$ before iteration because after shuffling, the probability of "$\mu_{\text{GDP}}(x_j^-, \delta_{\mathcal{A}}(x_j^+)) < \mu_{\text{GDP}}(x^-, \delta_{\mathcal{A}}(x^+))$ for all $j \in \{0, \cdots, k\}$" will be $\frac{1}{k+1}$. The expected occurrence of case 1 will be $\sum_{k=0}^{n+1} \frac{1}{k+1} = O(\log(n))$.

The time complexity of shuffling $\mathcal{S}$ is $O(n) = O(\epsilon_h c)$. Each binary search has a time complexity of $O(\log(c))$ and the expected number of binary searches is $O(\log(\epsilon_h c))$. The overall time complexity of the optimized algorithm is therefore $O(\epsilon_h c + \log(c)\log(c\epsilon_h)) = O(\epsilon_h c)$.

# E   Appendix: Plots

## E.1   The Laplace mechanism under GDP

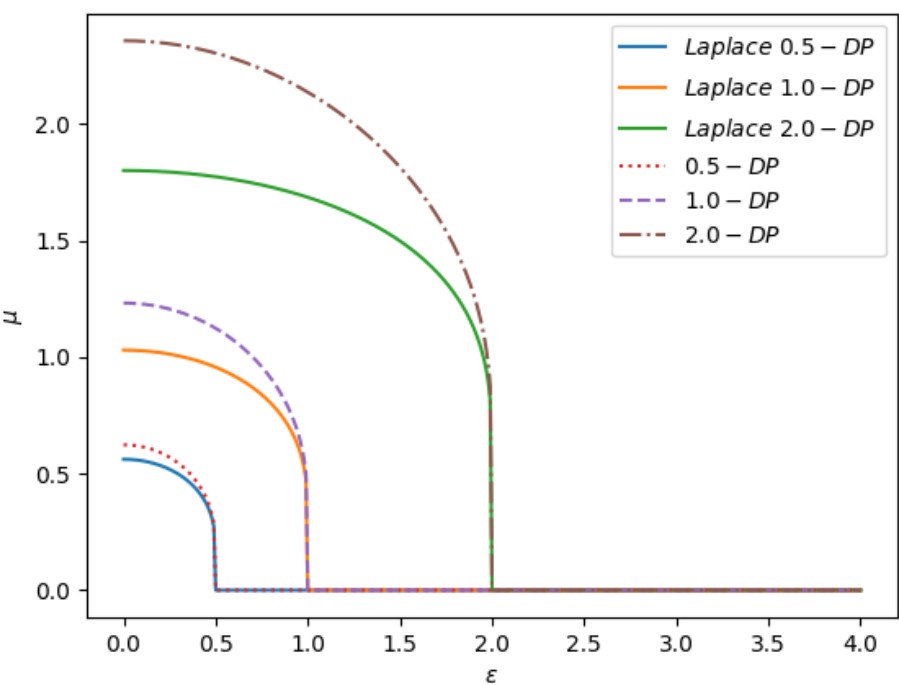

Figure 4: The plot of GDPT of $\epsilon$-DP privacy profiles and the Laplace mechanisms with the same $\epsilon$-DP guarantee. From the figure we can see the privacy protection provided by the Laplace mechanisms is slightly better than $\epsilon$-DP.

## E.2 The effect of subsampling

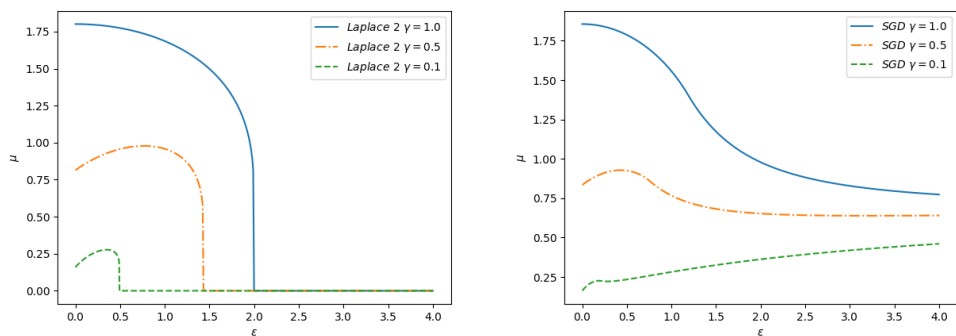

Figure 5: (Left) GDPT of the Laplace mechanism for various of $\gamma$. (Right) GDPT of the SGD for various of $\gamma$.

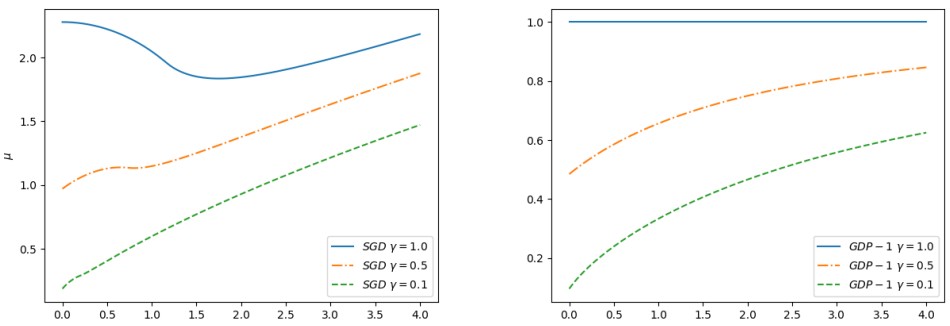

Figure 6: (Left) GDPT of the ICEA for various of $\gamma$. (Right) GDPT of the $\delta_\mu$ for various of $\gamma$. The Poisson subsampling procedure can significantly decrease the value of $\mu$ around $\epsilon = 0$ but has little effect on the GDPT's tail.