# OpenReview forum: "Identification, Amplification and Measurement: A bridge to Gaussian Differential Privacy"
_NeurIPS.cc/2022/Conference — NeurIPS 2022 Accept_

### Official Review · Reviewer_ipZH · 2022-07-10

**Rating:** 7
**Confidence:** 3
**Soundness:** 3 good
**Presentation:** 3 good
**Contribution:** 3 good

**Summary:**

This work shows the relationship between the common DP and GDP metric in privacy. GDP's good interpretability makes it clear than traditional DP with privacy profiles. The work follows the equivalence of GDP and DP with function $\delta_{\epsilon}$ and specifies the function characteristics to bridge DP and GDP. Besides the identification of DP to GDP, a  GDPT method is proposed to clearly show the relationship between DP and GDP in combination with head and tail analysis. The GDPT can also be that can be easily estimated supported by the theoretical analysis in the work. Finally, the authors show some applications in privacy compositions with the understanding of the GDP and DP.

**Questions:**

Some questions:
1. How is the cost/utility of GDP compared with other mechanisms? Some theoretical/magnitude results can be provided if possible.
2. Can GDPT help us design a new privacy mechanism with lower cost/ utility?
3. For the other algorithm with the privacy profile tail larger than $e^{-\delta^2}$, can we measure it with GDP approximately?
4. How is the proposed framework related to the privacy loss random variable? According to GDPT, $\mu$ gives a uniform description for the structure of the output's measurable space. The distribution/ the moments of privacy loss random variable might provide the detailed description, which may help the specification of the tail of GDPT.
5. Can Theorem 2.1 be stronger in dependence on actual algorithms? It may make the following analysis more concise.
6. In Theorem 4.3, is $x_i\in[0,\epsilon_h]$ or not?

Some suggestions:
1. Some detailed settings for the experiments should be shown. (Like dataset,or other settings)
2. It is better to restate the subsampling process to make it friendly to readers.
3. Some intuitive examples/ figures can be provided to show the tail larger than $e^{-\delta^2}$ can not be $\mu-$GDP.

**Ethics Review Area:**

["I don’t know"]

**Limitations:**

The authors should mention some limitations for heavy tails and head $\epsilon_h$ choices in the main context more clearly.

**Strengths And Weaknesses:**

Pros:
1. The work gives a powerful tool to bridge GDP and DP. It clearly reveals the relationship and provides a better understanding of GDP, which can make traditional DP more interpretable through GDP.
2. The good organization of the paper makes the readers easy to understand the proposed structure and corollaries.

Cons:
1. The practical application of DP to GDP is not clear. Can the proposed mechanisms help us to design a better privacy mechanism? Or how to quantify the interpretability?  How to show more effectiveness for GDP compared with DP?
2. The technical challenges/novelty should be stated clearly. The work is based on Theorem 3.1 a lot but the it is from [Dong et al. 2021].

---

> ### Author Response · Authors · 2022-08-02
> **Response**
>
> We sincerely appreciate the efforts the reviewer made and the positive feedback on our work. Below, we address your concerns. Please kindly let us know if there are any further questions.
>
> ### Novelty and motivation
>
> The research of DP can be traced back to Cynthia Dwork’s work in 2006 and the theory of GDP wasn’t formulated until 2019. The concept of GDP brings simple representation, exact composition theorems, and great interpretability. In addition, the central limit theorem in [13] shows that a sufficiently complex composition of DP-algorithms will have its privacy guarantee converge to GDP. This property hints that the GDP conditions might be very suitable for today’s increasingly complicated data-driven systems that need privacy protection. The development of DP in the past decade has given the community an abundance of algorithms and perturbed datasets but we are frustrated by the fact that those legacies aren’t reaching their full potential to make use of the advantages of GDP. The difficulties lie in the form of the privacy guarantee. Under the $(\epsilon,\delta)$-DP framework, the DP-guarantee can be falsely ruled out from GDP due to problematic asymptotic analysis (discussed in line 128-133 and 181-192 in our paper). That motivated us to conduct a further analysis and conclude that any algorithm that has a good enough tail in its privacy profile (provided or derived) is indeed qualified for GDP. After the binary "GDP or not" question is answered, we are interested in finding the value of $\mu$, which leads to theorem 4.2 and 4.3. The value $\mu$ can be then safely found due to bounded derivatives. Lastly, for algorithms that are disqualified from GDP, we gave a "clip and rectify" procedure (theorem 4.4) that can bring them back to the framework of GDP. The [Dong, 2019] built the theoretical foundation for GDP but left many questions unanswered. The gap between algorithms not developed under the GDP framework lies on both sides. For example, the privacy profile of GDP was derived, but we still need a practical numerical method to check for GDP conditions. Also, many old algorithms didn't give a full privacy profile, making the direct conversion to GDP very hard. The composition property discussed is also asymptotic and lacks the key tractable property for DP. Our work is aimed at mending those gaps.
>
> ### How is the cost/utility of GDP compared with other mechanisms?
>
> The advantage of GDP is beyond the scope of this paper, as our goal is to mend the gap instead of inventing GDP itself. However, we can still see the advantage of GDP in the section 5.2 as it shows a tigher privacy profile.
>
> ### Can GDPT help us design a new privacy mechanism ?
>
> Potentially yes. This is an interesting question that is not fully covered in this paper. In our paper, we considered a "clip and rectify" procedure to deal with algorithms with a very bad tail in their privacy profile. A bad tail in a privacy profile is the same as a large value of GDPT for larger $\epsilon$. In the appendix, we see that the subsampling has an effect on the head of GDPT. Although adding those procedures is not the same as designing a new mechanism, the GDPT can at least help us tweak some of the mechanisms we already have.
>
> ### Question: For the other algorithm with the privacy profile larger tail, can we measure it with GDP approximately?
>
> Yes. Such an algorithm can be measured by the head GDP conditions. Unlike the true GDP-conditions, a head-GDP condition always holds because any algorithm providing some non-trivial $(\epsilon,\delta)-$DP guarantee is $(0,\delta)$-DP for some $\delta<1$ (by theorem 2.1). In section 4.2 (line 259), we gave a  discussion about the gap between the head GDP and the true GDP. The users can assess the gap and make related decisions (e.g., acknowledge the extra risk or apply the rectify procedure).
>
> ### Question: How is the proposed framework related to the privacy loss random variable?
>
> It is known that the distribution of the privacy loss random variable is equivalent to the privacy profile . While the privacy loss random itself is a useful tool in the analysis of privacy conditions, it will not further enrich the information beyond the privacy profile.
>
> ### Question: Can Theorem 2.1 be stronger in dependence on actual algorithms?
>
> The Theorem 2.1 is tight in the sense that it can not be stronger for all $(\epsilon,\delta)$-DP algorithms in general. The fact that $(\epsilon,\delta)$-DP indicates $(\epsilon',\delta')$-DP for a slightly smaller $\epsilon'$ and slightly larger $\delta'$ means that privacy profiles can not decrease arbitrarily fast.  The "sufficiency" part of the proof is independent of any specific algorithms as it is purely based on the calculation of probabilities. We used an algorithm only to show the "necessity", which means the bound can not be improved.
>
> ### Question: In Theorem 4.3, is $x_i\leq \epsilon_h$ or not?
>
> Yes. We fixed the part to avoid this confusion.

---

### Official Review · Reviewer_GwbN · 2022-07-11

**Rating:** 5
**Confidence:** 3
**Soundness:** 2 fair
**Presentation:** 3 good
**Contribution:** 2 fair

**Summary:**

Gaussian DP is known as an exact privacy accountant with hypothesis-testing-based interpretation.
The paper studies a significant problem --- converting algorithms developed under other DP accountants into the GDP framework.
Specifically,
1. The authors study the connection between GDP and the privacy profile, and use the asymptotic properties of privacy profiles to identify the GDP property.

2. For non-GDP algorithms, the authors propose the ``clip and rectify'' procedure to convert these algorithms into a mu-GDP algorithm.

**Questions:**

Theorem3.3 is about the asymptotic property. It only provides a lower bound of GDP, but the derived GDP may not be as tight as the original privacy profile. It would be more interesting if the authors could plot both the original privacy profile and the converted GDP in a trade-off figure and compare the two curves.

**Ethics Review Area:**

["I don’t know"]

**Limitations:**

Yes, they have addressed it.

**Strengths And Weaknesses:**

Limitations: Though the conversion to GDP is solid,
the paper lacks empirical comparison or theoretical evidence to demonstrate that such conversion is tight. Other DP accountants might provide tighter privacy characterization if the conversion is loose.


Comments:

1. ``we show that all epsilon-DP algorithms are also GDP'' might be overclaimed. [13] has shown that the trade-off function of epsilon-DP will be lower bounded by the randomized response mechanism, which satisfies GDP.

2. The section 5.2 (handling composition with GDP) is not the contribution of this paper. It will be better to move it to the preliminary/related work section.

---

> ### Author Response · Authors · 2022-08-02
> **Response**
>
> Thank you for the overall positive feedback and valuable comments on our paper. Below, we address your concerns. Please kindly let us know if there are any further questions.
>
> ### Question: Theorem3.3 is about the asymptotic property. It only provides a lower bound of GDP, but the derived GDP may not be as tight as the original privacy profile.
>
> The purpose of this Theorem is to provide a lower bound of $\mu$ or an upper bound of privacy protection. The arbitrarily precise $\mu$ needs to be derived by algorithm 3. The $\mu$ derived in that way in the sense that it is arbitrarily close to the actual GDP level it provides.
>
> ### Comment: The convert can be lossy
>
> The $\mu$ derived in that way in the sense that it is arbitrarily close to the actual GDP level it provides. Like other parametric privacy condition families, the summarised value is not as comprehensive as the original curve. In our case, the value of $\mu$ will be conservative, especially for algorithms with particularly bad tails or heads. The GDP framework is not unversially suitable without modification, and our method is not limited to pure-DP of a certain family of algorithms. The central limit theorem in [13] shows that a sufficiently complex composition of DP-algorithms will have its privacy guarantee converge to GDP. This property hints that the GDP conditions might be very suitable for many data-driven systems that are becoming increasingly complicated today. In section 5.2, we conducted an experiment by converting the pure-DP to GDP and then applying the exact composition. This example has shown that after the lossy convert, the result is still better than the widely used advanced composition theorem. This is an interesting result as the conversion of a single pure-DP is not the optimal case (e.g. gaussian noise or after composition) considering the value of $\mu$ is bottlenecked by the privacy profile near the origin (Figure. 4). We want to express our gratitude for this comment of yours as it inspired us to revise the experiment in 5.2 by:
>
> 1. Derived the full curve of the optimal composition to give a more fair and complete comparison in the table.
>
> 2. Added a setting where the Theoerm 4.2 and 4.3 are applied after the composition to give a summary of the privacy profile using GDP (convert to GDP after the composition instead of before).
>
> We found that the "convert to GDP after composition" gives $1.420$-GDP which is a minor improvement over the $1.771$-GDP we had through this lossy conversion. For comparison, the central limit theorem  gives $1.414$-DP and this value is a lower bound due to an unknown gap between finite and infinite composition.
>
> ### Comment: ``we show that all epsilon-DP algorithms are also GDP'' might be overclaimed
>
> We have shown that GDP is a weaker tail condition of the privacy profile compared to pure-DP and conclude that all epsilon-DP algorithms are also GDP. This inclusion can be vaguely hinted by other literatures and we didn’t find anything that is explicitly in our form. In addition, we found the application of GDPT that shows the value of $\mu$ corresponding to $\mu$ quite interesting. That’s why we included this example here. We are happy to change the wording to "we show that the GDPT can give a value of $\mu$ for all pure-DP algorithms" if you still have concern that it is an overclaim.
>
> ### Comment: The section 5.2 (handling composition with GDP) is not the contribution of this paper.
>
> The experiment conducted here is not intended to replace the current study of compositions. We handled the composition with GDP to show that even with the slightly lossy conversion, the exact composition property of the GDP made the conversion worthwhile as it still performs better than the widely used "advanced" composition theorem. Our results are $\mu=1.771$ for conversion before composition and $\mu=1.420$ for composition before conversion, and the results are not asymptotic. The handling of composition with f-DP in [13] gives $\mu=\sqrt{2}$ but this value is a lower bound due to an unknown gap between finite and infinite composition.

---

### Official Review · Reviewer_vcSa · 2022-07-11

**Rating:** 6
**Confidence:** 3
**Soundness:** 4 excellent
**Presentation:** 3 good
**Contribution:** 3 good

**Summary:**

The authors in this paper extend the study of Gaussian differential privacy, an intuitive, single parameter privacy guarantee that exactly captures the privacy of the Gaussian mechanism. The authors introduce head and tail GDP, and provide conditions under which algorithms with given privacy curves/profiles satisfy these modes of GDP. The authors then show that algorithms, such as the Laplace mechanism, that satisfy pure DP also satisfy Gaussian DP. They demonstrate computationally that, by casting these algorithms under GDP, one can obtain tighter composition results than basic or advanced composition would permit.

**Questions:**

Nothing additional.

**Limitations:**

The authors say that the limitations are given as preconditions of theorems. It would helpful to the reader to also have the limitations separately addressed in conclusion, for instance, as this could motivate future work. For instance, one limitation is that the conversion to GDP is inherently lossy, and thus for many composition problems Fourier-based accounts can actually outperform the results in this paper, approximating the true privacy profile/curve to arbitrary precision.

**Strengths And Weaknesses:**

Strengths: Gaussian differential privacy is a simple and important concept for characterizing private algorithms. The authors demonstrate that, by furthering the study of GDP, one can obtain improved composition results for commonly used private algorithms. The authors also give computational techniques for characterizing the head GDP of algorithms, making the work practical. The work is original as it is the first to cast purely differentially private algorithms as GDP algorithms --- an approach that yields tighter composition results.

Weaknesses: Many of the algorithms considered in this work can have their exact privacy curve efficiently approximated using Fourier-based methods. While the results surrounding GDP offer improvements over baselines such as advanced composition, converting pure DP to GDP is still lossy. Since the paper contains many technical results, it may also be helpful to either provide sketches or explain intuitions to the reader.

---

> ### Author Response · Authors · 2022-08-02
> **Response**
>
> We sincerely appreciate the efforts the reviewer made and the valuable suggestions for our work.
>
> ### About the lossy conversion
>
> The $\mu$ derived in that way in the sense that it is arbitrarily close to the actual GDP level it provides. Like other parametric privacy condition families, the summarised value is not as comprehensive as the original curve. In our case, the value of $\mu$ will be conservative, especially for algorithms with particularly bad tails or heads. The GDP framework is not unversially suitable without modification, and our method is not limited to pure-DP of a certain family of algorithms. The central limit theorem in [13] shows that a sufficiently complex composition of DP-algorithms will have its privacy guarantee converge to GDP. This property hints that the GDP conditions might be very suitable for many data-driven systems that are becoming increasingly complicated today. In section 5.2, we conducted an experiment by converting the pure-DP to GDP and then applying the exact composition. This example has shown that after the lossy convert, the result is still better than the widely used advanced composition theorem. This is an interesting result as the conversion of a single pure-DP is not the optimal case (e.g. gaussian noise or after composition) considering the value of $\mu$ is bottlenecked by the privacy profile near the origin (Figure. 4). We want to express our gratitude for this comment of yours (and another reviewer) as it inspired us to revise the experiment in 5.2 by:
>
> 1. Derived the full curve of the optimal composition to give a more fair and complete comparison in the table.
>
> 2. Add a setting where the Theoerm 4.2 and 4.3 are applied after the composition to give a summary of the privacy profile using GDP (convert to GDP after the composition instead of before).
>
> We found that the "convert to GDP after composition" gives $1.420$-GDP which is a minor improvement over the $1.771$-GDP we had through this lossy conversion. For comparison, the central limit theorem  gives $1.414$-DP and this value is a only lower bound due to an unknown gap between finite and infinite composition.
>
> Please kindly let us know if there are any further questions.

---

### Official Review · Reviewer_cu8x · 2022-07-12

**Rating:** 6
**Confidence:** 2
**Soundness:** 2 fair
**Presentation:** 2 fair
**Contribution:** 2 fair

**Summary:**

The paper studies the asymptotic properties of privacy profiles and propose an efficient method for GDP algorithms to reduce the set of possible optimal privacy measurement. As an application to the main result, all pure-DP algorithms are also GDP.

**Questions:**

N/AN/

**Strengths And Weaknesses:**

Disclaimer: I was unable to read through the proof, but the results seems plausible. My comments are based on first 9 pages.

Strengths:
-- The paper presents analytic methods to deal with Gaussian DP, the one parameter privacy definition that captures entirety of approx-DP.
-- Gives an algorithm to find \mu with privacy profiles.


Weakness:
-- I found motivation of the paper and results a little bit difficult to understand.
-- There are couple of places where it is mentioned: nontrivial (\eps,\delta)-DP algorithm must be (0,\delta)-DP for some \delta <1. I really do not get this statement. Why is this true?
-- Which key lemma are the authors referring to on line 194.
-- I would suggest having the algorithm for finding \mu in the main 9 pages. It is very interesting algorithm.

---

> ### Author Response · Authors · 2022-08-02
> **Response**
>
> We sincerely appreciate the efforts the reviewer made and the valuable suggestions for our work. Below, we address your concerns. Please kindly let us know if there are any further questions.
>
> ### Motivation and results of this paper
>
> The research of DP can be traced back to Cynthia Dwork’s work in 2006 and the theory of GDP wasn’t formulated until 2019. The concept of GDP brings simple representation, exact composition theorems, and great interpretability. In addition, the central limit theorem in [13] shows that a sufficiently complex composition of DP-algorithms will have its privacy guarantee converge to GDP. This property hints that the GDP conditions might be very suitable for today’s increasingly complicated data-driven systems that need privacy protection. The development of DP in the past decade has given the community an abundance of algorithms and perturbed datasets, but we are frustrated by the fact that those legacies aren’t reaching their full potential to make use of the advantages of GDP. The difficulties lie in the form of the privacy guarantee. Under the $(\epsilon,\delta)$-DP framework, the DP-guarantee can be falsely ruled out from GDP due to problematic asymptotic analysis (discussed in line 128-133 and 181-192 in our paper). That motivated us to conduct a further analysis and conclude that any algorithm that has a good enough tail in its privacy profile (provided or derived) is indeed qualified for GDP. After the binary "GDP or not" question is answered, we are interested in finding the value of $\mu$, which leads to theorems 4.2 and 4.3. The value $\mu$ can be safely found due to bounded derivatives. Lastly, for algorithms that are disqualified from GDP, we gave a "clip and rectify" procedure (theorem 4.4) that can bring them back to the framework of GDP.
>
> ### Question: nontrivial ($\epsilon$,$\delta$)-DP algorithm must be (0,$\delta$)-DP for some $\delta <1$. I really do not get this statement. Why is this true?
>
> Answer: This is because of our theorem 2.1:
>
> The $(\epsilon_0$,$\delta_0)$-DP condition implies $(\epsilon,\delta)$-DP if and only if $\delta\geq \delta_0+ (1-\delta_0)(e^{\epsilon_0}-e^{\epsilon})^+/(1+e^{\epsilon_0})$. By plugging in $\epsilon=0$, we can see $(\epsilon,\delta)$-DP implies $(0,\delta)$-DP with $\delta=\delta_0+ (1-\delta_0)(e^{\epsilon_0}-1)/(1+e^{\epsilon_0})$. The term $(e^{\epsilon_0}-1)/(1+e^{\epsilon_0})$ is less than $1$. Therefore, the value of $\delta$ will be smaller than $1$ overall. We revised the paper to add a reference to Theorem 2.1 explicitly.
>
> ### Question: Which key lemma are the authors referring to on line 194?
>
> Answer: We are referring to Lemma 3.1 on line 193. We revised the paper to clearly indicate we are referring to the lemma above.
>
> ### Suggestion: I would suggest having the algorithm for finding $\mu$ in the main 9 pages. It is a very interesting algorithm.
>
> Response: We agree that algorithm 3 is interesting as it saves a lot of redundant calculations through random shuffling. We are happy to include this algorithm into the final version of our paper, but currently we are unable to fit this algorithm into the main pages due to the unchanged page limit in the rebuttal stage.

---

> > ### Comment · Reviewer_cu8x · 2022-08-06
> > **Improving the score**
> >
> > Apologies to the authors for responding so late.
> >
> > I thank the authors for clarifying my questions. I understand that they cannot include Algorithm 3 in the first 9 pages, but I would love to see the importance of the algorithm mentioned upfront.
> >
> > I am also increasing my score because I think the motivation in the response and the authors' response to other reviewer's questions have answered some of my other questions I had post submission of my initial review.

---

> > > ### Author Response · Authors · 2022-08-07
> > > **Acknowledgment**
> > >
> > > Thank you very much. We are grateful for the time and effort you put into improving our work.

---

### Meta-Review · Area_Chair_AUaM · 2022-08-25

**Recommendation:** Accept
**Confidence:** Certain

**Metareview:**

This paper expands our understanding of a recent variant of differential privacy, called Gaussian differential privacy, and studies its relationship with the standard definition. The reviewers all agree that the results are both important and interesting, and support accepting the paper.

**Award:**

No

---

### Decision · Program_Chairs · 2022-09-14

Accept